# To Guide or Not to Guide: Sparse Transductive Guidance for Program Synthesis

## Abstract

Program synthesis faces the dual challenge of achieving high success rates while maintaining interpretability and generalization, motivating hybrid approaches that combine complementary learning paradigms. Integrating transductive methods, which provide strong predictive power by directly mapping inputs to outputs but lack interpretability, with inductive methods, which excel at producing explicit and interpretable programs, creates a new opportunity for programming-by-example. While recent work has explored this integration through transductive guidance, we show, that permanent transductive guidance can, and in practice does, mislead search by overriding inductive reasoning strategies that would otherwise succeed. To address this limitation, we introduce TIIPS, a novel framework that, for the first time, applies transductive assistance sparsely and selectively to inductive synthesis. TIIPS adopts a teacher-student paradigm, where guidance is provided selectively, activated only when inductive synthesis fails, thereby preserving the natural problem-solving capabilities of inductive approaches. Experiments on two standard programming-by-example domains (string and list manipulation) demonstrate that TIIPS outperforms related work, solving more tasks and producing more robust solutions, particularly under distribution shifts. These results show that the timing and extent of transductive guidance matter more than its mere presence, establishing them as key factors for robust, interpretable, and effective program synthesis.

## 1 Introduction

Program synthesis aims to automate the generation of programs from high-level specifications, enabling systems to produce executable code from minimal or abstract guidance (Solar-Lezama, 2008). Within this field, programming-by-example (PBE) represents a particularly intuitive paradigm: tasks are specified via Input-Output (I/O) examples rather than formal logical descriptions (Devlin et al., 2017). By allowing code to be generated directly from user-provided examples, PBE promises to improve accessibility, reliability, and efficiency in software development (Gulwani, 2011). A key challenge in PBE is that specifications are rarely complete, inevitably leaving room for interpretation (Gulwani et al., 2017). Systems must therefore generalize from sparse examples, emulating human-like abstraction and reasoning (Chollet, 2019; Chollet et al., 2024).

PBE approaches can be broadly categorized into two paradigms. Inductive methods generate explicit, executable programs from task specifications that can be applied to novel inputs. These methods offer interpretability and reusability but face fundamental challenges with Domain-Specific Languages (DSLs): if the DSL is too restricted, the target program may be inexpressible; if too permissive, the search space becomes intractably large (Devlin et al., 2017). Transductive methods do not generate programs explicitly but instead directly infer outputs from test inputs based on the provided specifications. These methods inherently embody the latent function itself, potentially offering advantages when explicit rule formulation is difficult (Li et al., 2024). Tasks from the RobustFill string manipulation domain (Devlin et al., 2017) illustrate that (a) some problems are naturally solved inductively, (b) others require transduction, and (c) many demand a combination of both (Figure 1).

Recent work has highlighted the complementarity of inductive and transductive approaches (Li et al., 2024), demonstrating that superior performance can be achieved by combining these paradigms in an

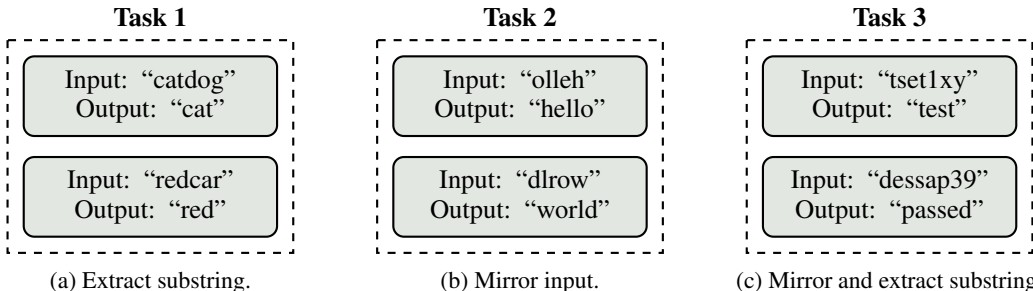

| Task 1 | Task 2 | Task 3 |
|---|---|---|
| Input: "catdog" Output: "cat" | Input: "olleh" Output: "hello" | Input: "tset1xy" Output: "test" |
| Input: "redcar" Output: "red" | Input: "dlrow" Output: "world" | Input: "dessap39" Output: "passed" |
| (a) Extract substring. | (b) Mirror input. | (c) Mirror and extract substring. |

Figure 1: Illustrative examples from the RobustFill (Devlin et al., 2017) string manipulation domain. (a) Certain tasks are naturally solved inductively, such as extracting a substring using the SubStr() primitive. (b) Other tasks, like mirroring an input string, cannot be solved inductively in a straightforward way because the RobustFill DSL lacks a reverse/mirror operator; while inductive solutions would require many operations, transductive models can directly learn the mapping. (c) Finally, some tasks require both paradigms illustrating how real-world problems often demand a combination of inductive rule application and transductive pattern recognition.

ensemble. However, in the approach by Li et al. (2024), both methods operate in isolation without interaction, and their outputs are simply aggregated rather than one method informing the other. While this demonstrates the value of combining paradigms, each approach faces distinct challenges that affect their practical applicability. Transductive approaches employ simpler learning pipelines, directly mimicking outputs without requiring explicit rule identification, making them suitable for handling irregular mappings (Kamnitsas et al., 2021). Yet, they produce black-box solutions that lack interpretability and may not generalize beyond their training context. In contrast, inductive approaches offer interpretability, enabling analysis, verification, and debugging of the generated program (Xu & Fekri, 2021), but require more complex learning pipelines to derive and encode underlying patterns, and face fundamental scalability challenges with respect to DSL design.

ExeDec (Shi et al., 2023b) can be viewed as a hybrid approach using transduction to enhance inductive program synthesis by providing structured guidance throughout the synthesis process. Essentially, it follows a divide-and-conquer principle: tasks are first decomposed into smaller subtasks by predicting the output of the next step, after which a program for the predicted subtask is generated. This procedure applies transductive guidance via task decomposition at *every step* of the inductive synthesis process, offering a systematic way to leverage the pattern recognition capabilities of transductive models while maintaining the interpretability benefits of inductive synthesis.

However, this *permanent* coupling between transductive and inductive approaches raises important questions about when and how such guidance should be applied. Consider an analogy: while a finance expert can provide valuable guidance to a data scientist developing a stock trading tool, the recommendations may not align with technical realities, potentially overlooking critical steps like data preprocessing or conflicting with system constraints such as memory limitations and API dependencies. Similarly, solving a (sub-)task transductively does not necessarily yield optimal guidance for solving it inductively. A transductive solution may overlook the specific reasoning patterns, intermediate steps, or structural insights that an inductive approach requires to succeed. This observation leads to our central hypothesis: while transductive guidance can be beneficial, overly rigid enforcement may actually hinder the natural problem-solving capabilities of inductive systems. Specifically, we hypothesize that *permanent* transductive guidance may override beneficial inductive reasoning processes, even in cases where the inductive approach alone would be more suitable for the given problem structure.

**Contributions.** In this work, we challenge the belief that permanent transductive guidance consistently improves inductive program synthesis. We first demonstrate through comprehensive ablation studies that permanent transductive guidance, while beneficial for certain tasks, can actually hinder the solution of problems that are better addressed through pure inductive reasoning. These findings extend previous work by revealing that transductively guided methods solve fundamentally different task subsets than purely inductive approaches, highlighting an important complementarity rather than universal superiority. Leveraging these insights, we propose Transductively Informed

Inductive Program Synthesis (TIIPS), a novel approach that applies transductive guidance only after inductive synthesis has failed and does so with incrementally increasing intensity. Specifically, TIIPS represents a paradigm shift from ExeDec's fully guided approach: rather than imposing guidance at every step, TIIPS begins with purely inductive synthesis and intervenes transductively only when needed, progressively increasing the degree of guidance until a solution is found. This incremental strategy ensures that problems are solved with the minimal necessary guidance, preserving the natural problem-solving capabilities of inductive methods while leveraging transductive support only when required. Empirical evaluation demonstrates that TIIPS consistently outperforms or matches ExeDec, producing more robust solutions and achieving superior performance across diverse generalization tasks. Overall, these results suggest that future hybrid approaches should focus on developing sophisticated methods to orchestrate the interplay between transductive and inductive reasoning, rather than simply maximizing the application of either paradigm.

## 2 BACKGROUND

### 2.1 COMPOSITIONAL GENERALIZATION

PBE allows users to specify programming tasks through I/O examples, bypassing the need for programming expertise (Gulwani, 2011). A PBE task is formally specified by a set of I/O pairs $(I_1, O_1), \ldots, (I_n, O_n)$ that capture the target program's intended functionality. We will refer to the entire set of task specifications as $\{(I_i, O_i)\}$. To formalize the solution search space, PBE typically operates within a DSL that defines the set of possible programs $P$. This DSL comprises functions, identifiers, constants, and variables as building blocks for program construction. The objective is to discover a program $p \in P$ where $p(I_i) = O_i$ for all $i \leq n$.

Compositional generalization refers to a model's ability to generalize to novel combinations of known components (Wiedemer et al., 2024). In the context of program synthesis, this capability can be systematically evaluated by constructing PBE tasks that induce specific distribution shifts between training and test data. Distinct task categories have been proposed (Shi et al., 2023b): 'Length generalization' tests handling of increased synthesis steps; 'compose different concepts' evaluates combining functions from separate categories; 'switch concept order' examines generalization to reversed function sequences; 'compose new operations' tests integration of isolated functions into compositions; and 'add operation functionality' assesses applying new functionalities to known operations. Details can be found in Appendix A.1 and Appendix A.2.

### 2.2 PERMANENT TRANSDUCTIVE GUIDANCE

ExeDec (Shi et al., 2023b) employs an iterative divide-and-conquer strategy to solve complex PBE tasks by decomposing them into smaller, manageable subtasks. A subtask is defined as a portion of the original task, also expressed in the form of I/O specifications, that can be solved with a single computational step–a DSL primitive and its arguments (e.g., `SubStr(0, 3)`). We refer to such a single computational step as a subprogram. ExeDec relies on two specialized Transformer models: a *Decomposition Model* that transductively predicts the output of the next subtask $\{\bar{O}_i\}$ given current task specifications $\{(I_i, O_i)\}$, and a *Synthesizer Model* that inductively generates subprograms mapping inputs to the predicted outputs.

The synthesis process follows a strict three-phase cycle that repeats until the task is solved or a step limit is reached. To illustrate, consider the task of abbreviating last names while keeping first names: (*"John Doe"*, *"John D."*), (*"Jane Smith"*, *"Jane S."*). ExeDec predicts the final output by sequentially predicting prefixes of it: each subtask output is a prefix of the target, and the concatenation of these predicted prefixes reconstructs the full solution over multiple steps.

(a) Decomposition: Given the current task specification, the Decomposition Model predicts what the outputs should be after the next computational step, forming a subtask specification. In our example, the model might predict intermediate outputs *"John"*, *"Jane"*, creating the subtask specification (*"John Doe"*, *"John"*), (*"Jane Smith"*, *"Jane"*).

(b) Synthesis: The Synthesizer Model receives this subtask specification and generates a subprogram that transforms the current inputs into the predicted outputs. For our subtask, it might generate `GetToken(WORD, 1)` to extract the first word from each input string.

(c) Updates: The generated subprogram is executed on the current inputs, and the results are used to update the task specification for the next iteration. After executing `GetToken(WORD, 1)`, the specification becomes (*"John Doe"*, *"D."*), (*"Jane Smith"*, *"S."*), which represents the remaining task.

This creates a *rigid transductive trajectory* where the Decomposition Model essentially attempts to solve the task step-by-step through output prediction, while the Synthesizer translates these predictions into executable programs. Both components are trained on datasets containing decomposed specifications and corresponding subprograms (details in Appendix A.3).

## 3 PURELY INDUCTIVE PROGRAM SYNTHESIS

We hypothesize that permanent transductive guidance as used in ExeDec can mislead program synthesis. To test this, we ablate ExeDec by removing its transductive Decomposition Model, creating a purely inductive baseline without any transductive guidance. This ablation allows us to isolate the effects of permanent transductive guidance and identify scenarios where it may be counterproductive.

### 3.1 PURELY INDUCTIVE ABLATION

To isolate its effect, we remove transductive guidance from ExeDec by omitting the (a) Decomposition step, yielding a purely inductive variant. While ExeDec decomposes tasks by predicting subtask outputs $\{(I_i, O_i)\}$ and guiding synthesis accordingly, the ablation directly generates subprograms from task specifications $\{(I_i, O_i)\}$ using only inductive synthesis. Both settings use the same synthesis model architecture ensuring comparability. This design allows us to evaluate systematically when transductive guidance helps and when it hinders program synthesis. Additional training information and algorithmic details are provided in Appendix B.1.

### 3.2 EXPERIMENTAL SETUP

**Domains.** We evaluate both approaches on the same two standard PBE benchmarks that are also used by ExeDec (Shi et al., 2023b): string (Devlin et al., 2017) and list manipulation (Balog et al., 2016), which are widely used in program synthesis (Nye et al., 2019; Hong et al., 2021; Jain et al., 2022). The RobustFill string manipulation focuses on transforming input strings into output strings using a DSL that provides operations such as substring extraction, modification, and composition. Programs in this domain typically consist of concatenations of largely independent expressions. For example, the program `Compose(ToCase(PROPER), GetFrom(' ')) | Const(',') | GetUpto(',')` transforms the input string "BRA, RIO DE JANEIRO" into "Rio De Janeiro, BRA".

DeepCoder list manipulation involves synthesis tasks over integer lists using a DSL that includes both first-order and higher-order functions such as `Map` and `Filter`. Programs are constructed line by line: each line applies an expression to either the original inputs or the result of a previous expression. For instance, the program `x0 = INPUT | x1 = Reverse x0 | x2 = ZipWith (+) x0 x1` transforms the input list $[12, 2, 13]$ into $[25, 4, 25]$. More detailed information on the domains, DSLs, and exemplary tasks can be found in Appendix A.1 and Appendix A.2.

**Evaluation and hyperparameters.** In ExeDec, a task is considered solved if the synthesized program produces the correct outputs for all $n$ provided I/O pairs. This evaluation does not assess whether the program generalizes to unseen inputs. In our setup, by contrast, we synthesize programs using only $n - 1$ I/O pairs and evaluate them on the held-out pair. A task is deemed solved only if the resulting program produces the correct output for both the unseen test input and all training inputs. Our experimental protocol follows ExeDec: a step limit of 10 for list tasks and 20 for string tasks, a beam size of 10, and 1,000 test tasks per category and seed, all using the same DSLs. The same pretrained models are used, with training details and hyperparameters provided in Appendix B.1. Results are reported as averages across 5 seeds, with 95% confidence intervals shown as error bars. Statistical significance between approaches is tested per category using paired t-tests. Significance levels are denoted by $*$ symbols (ns: $p \geq 0.05$, $*$: $p < 0.05$, $**$: $p < 0.01$, $***$: $p < 0.001$). Exact $p$- and $t$-values are provided in Appendix C.3.

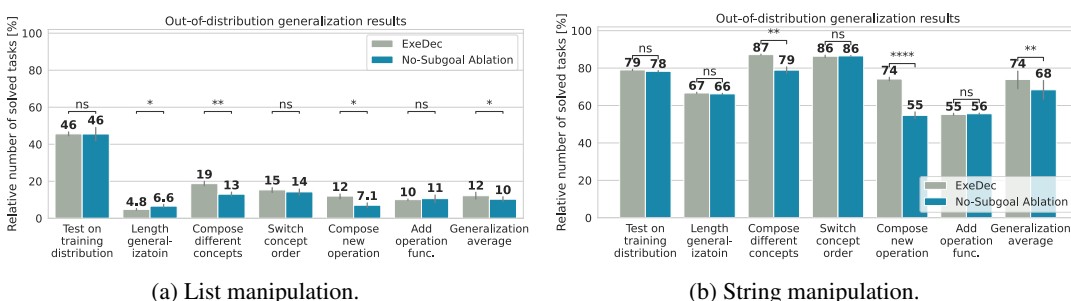

(a) List manipulation.      (b) String manipulation.

Figure 2: Performance of ExeDec compared to the Ablation evaluated on held-out test cases. The purely inductive ablation achieves a task coverage comparable to ExeDec.

### 3.3 Transductive Guidance Can Hurt Performance

To assess the impact of transductive guidance, we compare ExeDec to the purely inductive ablation. This ablation is architecturally identical to ExeDec, but it omits transductive guidance through explicit subtask decomposition. This setup mirrors the No-Subgoal Ablation from the ExeDec paper (Shi et al., 2023b), but evaluated on held-out test cases.

**Benefit of Permanent Transductive Guidance is Ambiguous.** As shown in Figure 2, both models solve a similar number of tasks across categories. In the string domain, ExeDec achieves an average generalization accuracy of $73.92\% \pm 12.29\%$, compared to $68.40\% \pm 12.92\%$ for the ablation. In the list domain, the gap is even smaller ($12.19\% \pm 4.97\%$ vs. $10.01\% \pm 3.61\%$) though statistically significant. ExeDec consistently outperforms the ablation in Compose Different Concepts and Compose New Operation, while the ablation performs better in Length Generalization within the list manipulation domain. In the remaining categories, the two approaches perform on par. As both differ only in the use of transductive guidance, these findings indicate that permanent transductive guidance in the form of explicit and permanent subtask decomposition provides no systematic benefit over purely inductive synthesis.

**Inductive and Transductively Guided Inductive Syntheses are Complementary.** To better understand the strengths of either approach, we analyze how many tasks are solved exclusively by either approach. For this purpose, we assign each solved task to the approach that solved it. Across all generalization categories in the list domain, ExeDec solves $43.16\%$ of tasks exclusively, the ablation solves $21.36\%$ exclusively, and both succeed on $35.47\%$. In the string domain, ExeDec solves $11.71\%$ exclusively, the ablation $4.56\%$, and both succeed on $83.72\%$. This demonstrates that each approach is capable of solving a substantial number of tasks that the other fails on, making them highly complementary. This observation aligns with findings from Li et al. (2024), who reported complementarity between purely inductive and purely transductive methods on the Abstraction and Reasoning benchmark (Chollet et al., 2024). Our results extend this conclusion to the interplay between inductive and transductively guided, inductive approaches. Furthermore, we repeat this ablation study using the same Synthesizer Model as in ExeDec, which was trained to generate subprograms from *subtask* specifications. Detailed experiments can be found in Appendix B.2 but the results confirm our previous findings: permanent transductive guidance can override inductive reasoning, enabling the solution of tasks that purely inductive approaches fail to solve, but at the cost of missing tasks that the purely inductive model can handle. This raises the key question of when transductive guidance is essential for solving a task and when it instead interferes with the natural problem-solving ability of inductive synthesis.

### 4 Transductively Informed Inductive Program Synthesis

The ablation study revealed an important insight: while transductive guidance can be crucial for solving certain tasks, it can also hinder succeeding in others. Specifically, permanent transductive guidance, as implemented in ExeDec, forces the model to solve tasks in a transductively predetermined way, which can overwrite otherwise correct inductive reasoning. At the same time, trans-

ductive guidance can push inductive synthesis to solve tasks that would otherwise remain unsolved. These findings demonstrate that we should not blindly apply transductive guidance at every step, but rather develop methods that leverage the complementary strengths of both paradigms more intelligently. Building on this observation, we introduce TIIPS, a framework that applies transductive guidance sparsely and only when needed. Unlike ExeDec, which enforces permanent subtask decomposition, TIIPS activates guidance only when the inductive synthesis loop fails to produce a solution and then increases the degree of guidance incrementally.

## 4.1 TIIPS FRAMEWORK

TIIPS operates through two interacting loops (Figure 3), both iterating for a maximum of $K = T$ steps. The inner loop performs stepwise inductive synthesis, mirroring the purely inductive ablation algorithm: given the current task specification $\{(I_i, O_i)\}$, the inductive synthesis model generates a subprogram. Next, the subprogram is executed on the current inputs yielding $\{\hat{O}_i\}$. If the program succeeds in producing the desired outputs $\{O_i\}$, a solution candidate is found. If not, the execution results are used to update the task specifications before the next step is synthesized. This cycle repeats for a maximum of $K$ iterations.

If no solution candidate is found after $K$ iterations, the outer loop introduces incremental transductive guidance. Initially, the transductive model is invoked to predict the output of the first subtask $\{\bar{O}_i\}$, which is then converted into a subtask specification. The inductive model uses the guidance for the first step to attempt the task, while the remaining steps are generated without transductive interference. If the task remains unsolved, the transductive model provides guidance for the first two subtasks, and the inductive model retries the task. This process continues incrementally, with the number of guided steps increasing on each iteration, until the task is successfully solved or the maximum number of outer iterations $T$ is reached. The first outer iteration corresponds to the purely inductive ablation algorithm, while intermediate iterations gradually increase the degree of transductive guidance. At the final outer iteration, TIIPS applies transductive guidance to all $T = K$ subprograms, reducing to ExeDec's permanent guidance.

TIIPS uses the same models as ExeDec yet shifts the guidance paradigm: it applies guidance selectively, only when inductive synthesis fails, and incrementally increases the number of guided steps. By avoiding permanent constraints, it preserves flexibility and prevents transductive guidance from overriding correct inductive solutions. This approach represents a first step toward more sophisticated methods for orchestrating the interplay between transductive and inductive reasoning, opening up a promising research direction for developing even more intelligent hybrid synthesis approaches. Additional algorithmic details can be found in Appendix C.1. A worked example of this selective, stepwise workflow is provided in Appendix C.2, contrasting TIIPS with ExeDec's permanent guidance.

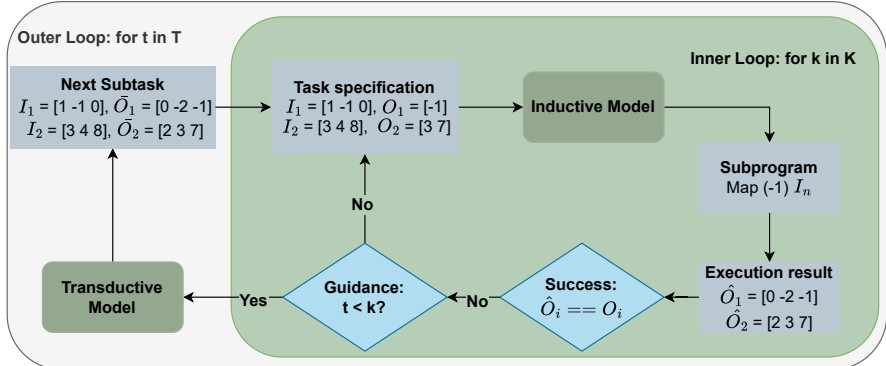

Figure 3: The TIIPS algorithm with sparse, on-demand transductive guidance. The inner loop mirrors the inductive synthesis process of the ablation model trying to synthesize a program. When the inductive process fails to produce a solution in $K$ steps, the outer loop incrementally introduces transductive guidance: Guiding the first step only, guiding the first two steps, ... .

## 4.2 IT IS ABOUT WHEN TO GUIDE AND NOT GUIDANCE PER SE

The central question of this study is not whether transductive guidance is helpful in general, but when it should be applied. We hypothesize that permanent guidance, as implemented in ExeDec, pushes inductive synthesis toward solving certain tasks but often at the cost of failing to solve others that could be solved without guidance. Conversely, we hypothesize that TIIPS, which applies transductive guidance only when necessary and increases its degree incrementally, strikes a better balance between inductive reasoning and transductive support.

**Evaluation.** We evaluate TIIPS on the same domains, use the same hyperparameters, random seeds, and experimental settings as described in the Section 3.2. Both the inner and outer loops are limited to 10 steps for the list and 20 steps for the string domain. Due to the nested loop structure of TIIPS, it can generate up to 1000 programs in the list domain and 4000 programs in the string domain. For a fair comparison, ExeDec is given 10 attempts per task in the list and 20 attempts per task in string manipulation domain thus matching the total number of evaluated programs by TIIPS. This ensures compute alignment in that all approaches can generate the same number of programs.

**Sparse & Selective Transductive Guidance Boosts Performance.** Figure 4 reports the relative number of solved tasks in both domains. In terms of average generalization, TIIPS demonstrates a significant advantage, solving on average $20.17\% \pm 8.14\%$ of tasks compared to ExeDec's $12.28\% \pm 5.01\%$ in the list domain. In the string domain, both approaches perform on par, with ExeDec reaching $73.92\% \pm 12.29\%$ on average and TIIPS achieving $73.01\% \pm 12.20\%$. These results highlight that sparse, selective guidance preserves the natural problem-solving ability of inductive synthesis while retaining the benefits of transductive guidance. A deeper look at task overlaps further clarifies this picture (Figure 5). In the list domain, TIIPS solves more than $40\%$ of all solved tasks that are not solved by ExeDec and accounts for $11.78\%$ of all solved tasks exclusively. By contrast, ExeDec solves only about $3.5\%$ of solved tasks that TIIPS cannot solve. Tasks that TIIPS fails to solve but ExeDec succeeds on can be attributed to the last step not being guided: TIIPS may find a program that solves all training I/O pairs except the test pair, producing a false-positive program. In the string domain, the differences are minimal: ExeDec covers roughly $1.5\%$ of solved tasks exclusively, while TIIPS contributes only $0.3\%$. Tasks that TIIPS fails to solve but the ablation succeeds are likely due to the matched compute budget for the ablation, which allows multiple synthesis attempts per task. These results suggest that in domains where permanent guidance is beneficial, TIIPS can still exploit the potential of permanent transductive guidance, while remaining dynamic enough to fall back on inductive reasoning or intermediate levels of guidance in domains where permanent guidance becomes harmful.

**Sparse & Selective Transductive Guidance Generates More Robust Programs.** Beyond quantitative performance analysis, we also examine the robustness of generated solutions using density plots (Figure 6). Each solved task in the list domain is positioned according to two measures: intent match on the x-axis and syntactical overlap on the y-axis. Intent match quantifies semantic alignment by comparing the predicted subtasks or, in the case of unguided synthesis, the execution results of the predicted subprogram, with the ground-truth outputs, while syntactical overlap cap-

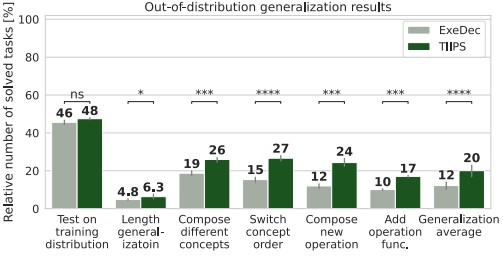

(a) List manipulation.

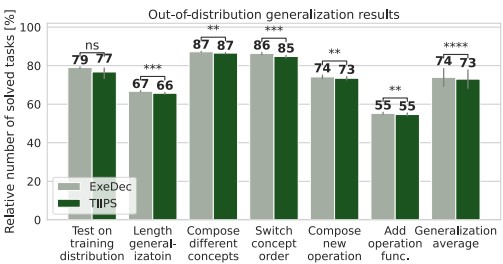

(b) String manipulation.

Figure 4: Performance of TIIPS compared to ExeDec. In the list domain, TIIPS outperforms ExeDec in all categories. In the string domain, both approaches achieve similar results.

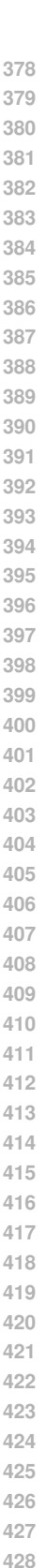

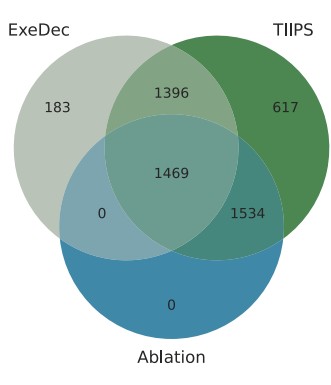 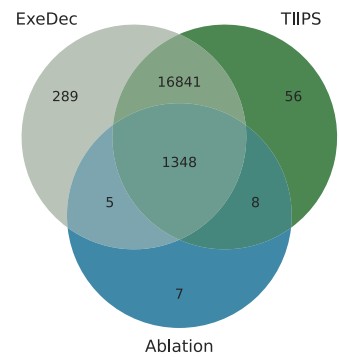

(a) List manipulation.                    (b) String manipulation.

Figure 5: Venn diagrams showing overlap of solved tasks across all categories and seeds between TIIPS, ExeDec, and the Ablation in both domains. In the list domain, TIIPS solves about 40% of tasks that are not solved by ExeDec. In the string domain, ExeDec solves slightly more tasks than TIIPS.

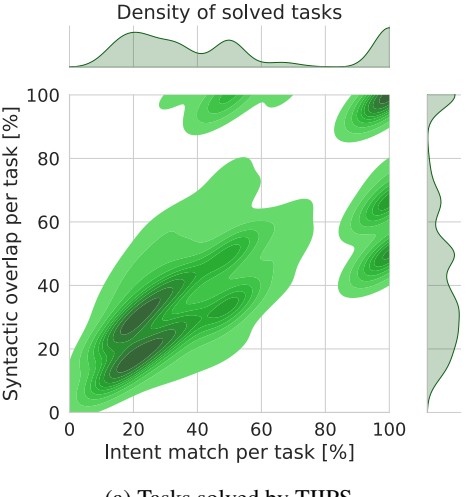 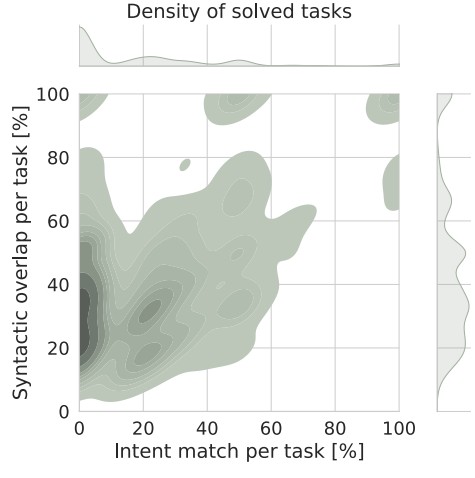

(a) Tasks solved by TIIPS.               (b) Tasks solved by ExeDec.

Figure 6: Kernel density plot of tasks solved by TIIPS and ExeDec in the list domain, grouped by intent match and syntactical overlap. Intent match measures the overlap between predicted/executed and ground-truth subtask outputs, while syntactical overlap reflects the structural similarity of predicted programs to ground-truth solutions. As a result, correctly solved tasks, both semantically and syntactically, concentrate in the top-right region. Results of both approaches are averaged across all seeds and compositional generalization categories.

tures structural similarity between predicted programs and ground-truth solutions. For example, in a four-step task where an approach predicts only three subtasks correctly and generates two syntactically aligned subprograms, the resulting data point would lie at (75%, 50%). When comparing the distributions in the list domain, TIIPS produces solutions that cluster more tightly in the top-right region than those of ExeDec. A similar tendency can be observed in the string domain (Figure 15), though the effect is less pronounced than in the list domain. This suggests that TIIPS not only solves a larger fraction of tasks but also produces solutions that better preserve both semantic intent and syntactic form.

## 5 RELATED WORK

**Programming-by-example.** Program synthesis from examples has been addressed through a variety of neural and symbolic methods. Neurally-guided synthesis approaches use learned models to prioritize program candidates during symbolic search (Balog et al., 2016; Yin & Neubig, 2017; Lee et al., 2018). Multi-step synthesis strategies apply search over partial programs, either in a top-down manner (Nye et al., 2019; Ellis et al., 2021) or through bottom-up enumeration (Odena et al., 2020; Shi et al., 2022; 2023a). Execution-guided synthesis incorporates intermediate execution results to inform program generation (Ellis et al., 2019; Chen et al., 2018; Shrivastava et al., 2021). More recently, planning-based methods have framed synthesis as a sequential decision-making problem, using structured search or latent planning to guide program construction (Murali et al., 2017; Nye et al., 2019; Chen et al., 2020; Hong et al., 2021; Klinger et al., 2023; Prasad et al., 2023; Zhang et al., 2023; Cano et al., 2023; Witt et al., 2023; Khan et al., 2025; Demirtaş et al., 2025).

**Hybrid Methods.** Combining inductive and transductive paradigms for PBE has been explored in multiple ways. Some approaches treat the two models as independent components, aggregating their predictions without iterative interaction (Li et al., 2024). While showing that induction and transduction can be complementary, their analysis only compares purely inductive and purely transductive models, without investigating interactions within transductively guided induction. ExeDec (Shi et al., 2023b) introduced transductive task decomposition to guide inductive synthesis. It follows a divide-and-conquer principle where complex tasks are broken into subtasks via transduction, but it applies transductive predictions at every position regardless of the inductive solver's progress, which can sometimes hinder inductive solutions. Finally, they show that this concept can be transferred to Large Language Models (LLMs). However, they consistently underperform relative to transformer models trained from scratch in these domains and tasks (Shi et al., 2023b). Last, the LLM-ARChitect system, integrates inductive and transductive components differently by using inductively derived data augmentation rules to generate additional examples for test-time training (Franzen et al., 2024).

## 6 CONCLUSION

This work reveals a fundamental tension in hybrid program synthesis: the very guidance mechanisms designed to enhance performance can paradoxically constrain the systems they aim to help. Our findings challenge the prevailing assumption that more guidance is inherently better, instead suggesting that the timing and intensity of guidance matter as much as its quality. The complementary nature of transductive and inductive approaches we uncovered has broader implications for AI system design. If two reasoning paradigms consistently solve disjoint problem sets, this suggests that current evaluation practices, which typically report aggregate performance, may obscure critical insights about when and why different approaches succeed or fail. This observation calls for more nuanced evaluation frameworks that can capture such complementarity patterns across AI domains.

Perhaps most significantly, our results point toward a new research paradigm: instead of asking "how can we provide better guidance?" we should be asking "when should we provide guidance?" The success of TIIPS's simple adaptive strategy suggests that even modest improvements in guidance orchestration can yield substantial benefits. This opens exciting avenues for learning dynamic guidance policies that could adapt not just to task difficulty, but to the specific reasoning patterns required by different problem structures.

**Limitations.** Our study focuses on DSL-based synthesis. While LLMs have shown impressive capabilities in PBE, DSL-based approaches remain competitive on structured tasks and offer greater interpretability for studying the interplay between reasoning paradigms (Shi et al., 2023b). Recent work demonstrating that DSL-constrained approaches can scale to complex domains (Li et al., 2024) suggests our findings remain relevant as the field evolves toward hybrid architectures that combine the strengths of both specialized and general models. TIIPS represents only a first step toward intelligent guidance orchestration. Its rigid incremental strategy always starting guidance from the first step highlights a key open problem: developing methods that can dynamically identify which specific steps truly benefit from intervention. Future work might explore learned guidance policies, multi-modal reasoning coordination, and adaptive systems that can recognize their own reasoning limitations in real-time.

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

# A BACKGROUND

## A.1 STRING MANIPULATION (ROBUSTFILL) DOMAIN

The string manipulation or RobustFill domain focuses on transforming input strings into output strings using an DSL composed of operations such as substring extraction, modification, and composition. Each example consists of a single input string, and the corresponding output is also a single string. (Devlin et al., 2017)

### A.1.1 DSL OPERATIONS AND PROGRAM STRUCTURE

Figure 7 shows all DSL operations used in this work. In the string manipulation domain, programs are structured as concatenations of subprograms, with the exception of the `Compose` operation that permits nesting of depth two. An example of such a program can be seen in Figure 8. For a given task specified as $\{(I_i, O_i)\}$, the decomposition procedure maintains an updated specification after each predicted subprogram. In this domain, the inputs $\{I_i\}$ remain unchanged across all steps, while the outputs are shortened by removing the portion of the string already explained by the current subprogram. Concretely, if the predicted subprogram applied to the input yields $\{\hat{O}_i\}$, and the current target output is $\{O_i\}$, then the new state becomes $\{(I_i, O_i - \hat{O}_i)\}$. For example, suppose the input is the string "foobar" and the ground-truth program is composed of two subprograms: one that extracts "foo" and another that extracts "bar". The target output "foobar" is first updated to "bar" after predicting the first subprogram which yields "foo", and then to the empty string after predicting the second. In this way, the residual output always represents the part of the task that still needs to be solved.

A key characteristic of this domain is how tasks are generated: subprograms are sampled at random and appended to form a complete program, which is then applied to a randomly chosen input. As a result, each task is inherently defined by the sequence of subprograms used to construct it, and therefore by the corresponding sequence of subtasks. This means that the decomposition pattern of a task is fixed or allows at most a very limited number of alternatives. Consider for instance the program `GetAll(NUMBER) | Const('.') | GetToken(WORD, -1)`. Here, the task requires first extracting all numbers from the input, then inserting a dot, and finally appending the last word yielding "1.Turning" for the input string "alan Turing1". There is no realistic way to permute or substitute the order of these subtasks: if the second and third operations were swapped, or if a different sequence of operations were attempted, the output would no longer match the target. The only conceivable alternative would be to decompose the last step in an artificially fine-grained way, for example, by producing the last word character by character rather than as a whole. Such

$$
\begin{array}{rl}
\text{Program } P := & \texttt{Concat}(e_1, e_2, \ldots) \\
\text{Expression } e := & s \mid m \mid o \mid \texttt{ConstStr}(c) \\
\text{Compose } o := & m_1(m_2) \mid m(s) \\
\text{Substring } s := & \texttt{SubStr}(k_1, k_2) \mid \texttt{GetSpan}(r_1, i_1, b_1, r_2, i_2, b_2) \\
& \mid \texttt{GetUpto}(r, i) \mid \texttt{GetFrom}(r, i) \mid \texttt{GetToken}(r, i) \\
\text{Modification } m := & \texttt{ToCase}(a) \mid \texttt{Replace}(c_1, c_2) \mid \texttt{Trim}() \\
& \mid \texttt{GetFirst}(r, i) \mid \texttt{GetAll}(r) \\
& \mid \texttt{Substitute}(r, i, c) \mid \texttt{SubstituteAll}(r, c) \\
& \mid \texttt{Remove}(r, i) \mid \texttt{RemoveAll}(r) \\
\text{Regex } r := & \texttt{NUMBER} \mid \texttt{WORD} \mid \texttt{ALPHANUM} \mid \texttt{ALL\_CAPS} \mid \texttt{PROPER\_CASE} \\
& \mid \texttt{LOWER} \mid \texttt{DIGIT} \mid \texttt{CHAR} \mid \delta \\
\text{Case } a := & \texttt{ALL\_CAPS} \mid \texttt{PROPER\_CASE} \mid \texttt{LOWER} \\
\text{Position } k := & -100 \mid -99 \mid \ldots \mid -1 \mid 0 \mid 1 \mid 2 \mid \ldots \mid 100 \\
\text{Index } i := & -5 \mid -4 \mid \ldots \mid -1 \mid 1 \mid 2 \mid \ldots \mid 5 \\
\text{Boundary } b := & \texttt{START} \mid \texttt{END} \\
\text{Character } c := & A \mid \ldots \mid Z \mid a \mid \ldots \mid z \mid 0 \mid \ldots \mid 9 \mid \delta \\
\text{Delimiter } \delta := & \texttt{\&} \texttt{,} \texttt{.} \texttt{?} \texttt{!} \texttt{@} \texttt{()} \texttt{[]} \texttt{\%} \texttt{\#} \texttt{\$} \texttt{"} \texttt{'}
\end{array}
$$

Figure 7: String manipulation primitives.

**String Manipulation Task**

---
**I/O Pair 1**
Input: `alan Turing1`
Output: `1.TURING,Alan`

---
**I/O Pair 2**
Input: `21.Donald@knuTh`
Output: `21.KNUTH,Donald`

---
**I/O Pair 3**
Input: `8:grace,HoppER`
Output: `8.HOPPER,Grace`

---
**I/O Pair 4**
Input: `EDSGER99 DIJKSTRA`
Output: `99.DIJKSTRA,Edsger`

---

```
Ground Truth: GetAll(NUMBER) | Const('.')  |
Compose(ToCase(ALL_CAPS), GetToken(WORD, -1)) |
Const(',') | Compose(ToCase(PROPER), GetToken(WORD, 1))
```

Figure 8: Exemplary task from the string manipulation domain. The task rearranges the input string so it starts with the number, followed by the last name in caps, and the first name in title case.

alternatives, however, are unnatural and not consistent with how tasks are generated. Thus, while the syntactic realization of each subprogram may vary, the decomposition structure itself is essentially predetermined. The only softening of this rigidity is provided by the `Compose` operation, which permits limited nesting and thus introduces a small degree of flexibility.

### A.1.2 BENCHMARK CREATION

In the string manipulation domain, programs consist of concatenated subprograms, with program length defined as the number of subprograms (Shi et al., 2023b). To evaluate compositional generalization, five benchmarks are used that test different aspects of a model's ability to generalize beyond the training distribution. These tasks can be grouped into three overarching themes: length generalization, mixing and matching concepts, and applying general principles.

- **Length Generalization:** This category measures whether a model can produce longer programs than those seen during training. Training programs have lengths 1–6, while test programs are longer, with lengths 7–10.

- **Compose Different Concepts:** This task evaluates whether a model can combine concepts in ways not seen during training. DSL operations are partitioned into "substring" and "non-substring" groups (excluding `Compose`). Training tasks contain operations from only one group, while test tasks combine operations from both groups. Program lengths range from 2–6 for both training and testing. This setup tests the model's ability to generalize to novel combinations of learned operations.

- **Switch Concept Order:** Here, the ability of a model to recombine concepts in sequences not seen during training is assessed. Training programs always place substring operations before non-substring operations, while test programs reverse this order. Program lengths for both training and testing are 2–6.

- **Compose New Operation:** This category examines the ability to integrate a new, previously isolated operation into larger compositions. One quarter of training tasks consist of single-op `Compose` programs, while the remaining training tasks (lengths 2–6) exclude

Compose. Test programs (lengths 2–6) include Compose. The goal is to test whether a model can leverage knowledge of an isolated operation and incorporate it into more complex compositions.

- **Add Operation Functionality:** This task tests whether a model can extend its understanding of an operation by inferring new functionality from context or analogies with other operations. Training tasks (lengths 1–6) omit certain behaviors of substring operations within Compose, while test programs include these behaviors. This reflects scenarios such as using updated library functions with new parameters that can be inferred from analogous existing operations.

### A.2 LIST MANIPULATION (DEEPCODER) DOMAIN

The list manipulation or DeepCoder domain (Balog et al., 2016) involves reasoning over integer lists using a DSL that supports both first-order and higher-order functions, such as Map and Filter. Tasks may include multiple input variables, each representing either integers or lists of integers, while the output is a single integer or a single integer list. Intermediate task specifications are derived by executing the current partial program on the input variables. The resulting execution output then serves as an input variable for the subsequent synthesis step. Contrary to the string manipulation domain, intermediate program states do not directly capture what is left to do. This information must be derived by comparing the current input variable and the overall target output.

#### A.2.1 DSL OPERATIONS AND PROGRAM STRUCTURE

Figure 9 shows all DSL operations used in the list domain. They are the same as used in the original ExeDec study (Shi et al., 2023b). Programs are constructed sequentially, with each line depending on the outputs of preceding expressions and the initial output variable (Figure 10). This structure mirrors human coding practices and enables opportunities for intermediate error correction. The domain's design results in a larger combinatorial space and supports more expressive decomposition strategies, thereby increasing the complexity of the synthesis process. A real example of a final synthesized program for a representative task is shown in Appendix A.2.

$$
\begin{aligned}
\text{Program } P :=\ & i_1;\ i_2;\ \ldots;\ a_1;\ a_2;\ \ldots \\
\text{Initialization } i :=\ & v \leftarrow \texttt{INPUT} \\
\text{Assignment } a :=\ & v \leftarrow f \mid v \leftarrow h \\
\text{First-Order Operation } f :=\ & \texttt{Head}(l) \mid \texttt{Last}(l) \mid \texttt{Access}(n,l) \mid \texttt{Minimum}(l) \mid \texttt{Maximum}(l) \\
& \mid \texttt{Sum}(l) \mid \texttt{Take}(n,l) \mid \texttt{Drop}(n,l) \mid \texttt{Reverse}(l) \mid \texttt{Sort}(l) \\
\text{Higher-Order Operation } h :=\ & \texttt{Map}(\lambda,l) \mid \texttt{Filter}(\beta,l) \mid \texttt{Count}(\beta,l) \mid \texttt{Zip}(\Sigma,l,l) \\
& \mid \texttt{Scanl1}(\Sigma,l) \\
\text{int} \rightarrow \text{int Lambda } \lambda :=\ & (+1) \mid (-1) \mid (*2) \mid (/2) \mid (*(-1)) \mid (**2) \mid (*3) \mid (/3) \mid (*4) \mid (/4) \\
\text{int} \rightarrow \text{bool Lambda } \beta :=\ & (>0) \mid (<0) \mid (\%2 == 0) \mid (\%2 == 1) \\
(\text{int}, \text{int}) \rightarrow \text{int Lambda } \Sigma :=\ & (+) \mid (-) \mid (*) \mid (\min) \mid (\max) \\
\text{Integer Variable } n :=\ & v \\
\text{List Variable } l :=\ & v \\
\text{Variable Name } v :=\ & x_1 \mid x_2 \mid \ldots
\end{aligned}
$$

Figure 9: First and Higher-Order Functions contained in the DSL for the list manipulation domain.

**List Manipulation Task**

**I/O Pair 1**
Input: `[5, 1, 6], [-2, -15, 1]`
Output: `[9, -30, 7]`

**I/O Pair 2**
Input: `[5], [-4]`
Output: `[3]`

**I/O Pair 3**
Input: `[10, 2], [-2, -5]`
Output: `[0, 15]`

Ground Truth:
$x_0 = $ `INPUT` $\mid x_1 = $ `INPUT` $\mid x_2 = $ `Scanl1 (max) ` $x_0$
$\mid x_3 = $ `ZipWith (+) x1 x2` $\mid x_4 = $ `Map (*3) ` $x_3$

Figure 10: Example from the list manipulation domain. The task requires to extract the running maximum of the first input, add this to the second input, and multiply the resulting list by 3.

### A.2.2 BENCHMARK CREATION

In the list manipulation domain, programs are structured line-by-line, with task length defined by the number of non-input lines (Shi et al., 2023b). In the DeepCoder list domain, each non-input line of a program is treated as a subprogram, so program length corresponds to the number of computational steps. Programs are evaluated using I/O examples, with at most two input lists. Five compositional generalization tasks are used that assess a model's ability to synthesize longer programs, recombine operations in new ways, and extend operation functionality, mirroring the meta-benchmark used in the string domain.

- **Length Generalization:** Training programs have lengths 1–4, while test programs are length 5. This task evaluates whether a model can generalize to longer programs and more complex compositions than seen during training.

- **Compose Different Concepts:** This task assesses the ability to combine operations from different conceptual groups. Training programs include operations from one of two concepts: (i) all first-order operations plus `Map`, and (ii) all remaining higher-order operations. Test programs combine operations from both concepts, with lengths 1–4, to test generalization to novel combinations.

- **Switch Concept Order:** Similar to Compose Different Concepts, this task tests whether a model can apply known concepts in a different order. Training programs follow one fixed sequence of operations across the two concepts, while test programs reverse the order. Program lengths range from 1–4 for both training and testing.

- **Compose New Operation:** This task evaluates the integration of a previously isolated operation into larger compositions. 25% of training tasks are length-1 programs containing only the `Scanl1` operation, while the remainder are length 2–4 programs that exclude `Scanl1`. Test programs are length 2–4 and include `Scanl1`, measuring whether the model can leverage isolated knowledge to compose more complex programs.

- **Add Operation Functionality:** This task assesses whether a model can extend its understanding of an operation by generalizing to previously unseen use cases. Training programs (lengths 1–4) use `Scanl1` only with the lambdas `(-)` and `(min)`, while test programs apply `Scanl1` with other lambdas `(+)`, `(*)`, and `(max)`. This evaluates the model's ability to infer new operation behaviors from existing examples.

## A.3 EXEDEC

ExeDec (Shi et al., 2023b) represents a neural divide-and-conquer approach to program synthesis that addresses the complexity of PBE tasks through iterative decomposition. The core insight underlying ExeDec is that complex programming tasks can be systematically broken down into sequences of simpler subtasks, each solvable by a single DSL primitive operation. ExeDec's architecture consists of two complementary neural components working in tandem: The transductive Decomposition Model which takes the current task specification and predicts what the outputs of the next computational step could be. The model operates transductively by reasoning about specific I/O transformations rather than learning general synthesis rules. The inductive Synthesizer Model takes the subtask specification as predicted by the Decomposition Model and generates the corresponding subprogram, i.e., a DSL primitive with its appropriate arguments. The goal of the Decomposition Model is to decompose the task into subtasks that are expedient and the goal of the Synthesizer Model is to predict subprograms that map the current inputs to the predicted intermediate outputs. The ExeDec algorithm (Algorithm 1) follows a rigid iterative procedure that enforces a transductive trajectory: At each iteration $k$, the Decomposition Model examines the current task state $\{(I_i, O_i)\}$ and predicts the intermediate outputs $\{\bar{O}_i\}$. This prediction effectively defines a subtask $\{(I_i, \bar{O}_i)\}$ that represents a single step toward the final solution. The Synthesizer Model takes this subtask specification and generates a subprogram. The generated subprogram is executed on all current inputs, producing actual outputs $\{\hat{O}_i\}$. If these outputs match the target outputs $\{O_i\}$, the algorithm terminates successfully with the current sequence of subprograms forming the complete solution. If the task is not yet solved, the algorithm updates the task specification based on the execution results. This update is domain-specific: In the list domain the execution results $\{\hat{O}_i\}$ become the new inputs, while target outputs remain unchanged, yielding state $\{(\hat{O}_i, O_i)\}$. In the string domain, the inputs remain unchanged, but target outputs are shortened by removing the portion already produced, resulting in state $\{(I_i, O_i - \hat{O}_i)\}$.

**Model Training.** Training data is generated through randomly sampling a program from the DSL and applying it to random inputs. Each program is a sequence of subprograms that are categorized as detailed in Appendices A.1.2 and A.2.2. For each subprogram, the training data incorporates three elements: (A) the program state resulting from executing preceding subprograms, (B) the execution result of the current subprogram, and (C) the subprogram itself. The Decomposition Model is trained to predict the execution result (B) of the current subprogram given the current program state (A). The Synthesizer Model learns to predict the subprogram (C) conditioned on subtask specifications–specifically, (A) augmented with (B). All models undergo separate training

---

**Algorithm 1** ExeDec algorithm using permanent transductive guidance.
Note, $\{x_i\}$ is short for $[x_1, \ldots, x_n]$, where $n$ is the number of I/O pairs.

---

**Require:** Inductive step limit $K$
**Ensure:** full_program
1: Initialize full_program $\leftarrow$ []
2: current_state $\leftarrow \{(I_i, O_i)\}$
3: **for** $k = 1$ to $K$ **do**
4:     subtask $\leftarrow$ DecompositionModel(current_state)        ▷ Decompose task into next subtask
5:     subprogram $\leftarrow$ SynthesizerModel(subtask)
6:     $\{\hat{O}_i\} \leftarrow$ Execute(subprogram, $\{I_i\}$)
7:     **if** $\{\hat{O}_i\} = \{O_i\}$ **then**                                        ▷ Found successful program
8:         full_program $\leftarrow$ full_program + subprogram
9:         **return** full_program
10:     **if** domain = "list" **then**                                  ▷ Domain-specific updates
11:         current_state $\leftarrow \{(\hat{O}_i, O_i)\}$
12:     **else if** domain = "string" **then**
13:         current_state $\leftarrow \{(I_i, O_i - \hat{O}_i)\}$
14:     full_program $\leftarrow$ full_program + subprogram
15: **return** full_program

---

for each generalization task, enabling specialization to the specific characteristics of individual task domains. The same architectures and hyperparameters as in the original ExeDec paper were used for both Decomposition and Synthesizer Model. The setup used an embedding dimension of 512, a hidden dimension of 1024, 3 layers, and 4 attention heads. For relative attention, 32 buckets for relative position embeddings, with logarithmically spaced bucket boundaries, were used. The maximum relative distance was determined based on the input and output sequence lengths. Models were trained using the Adam optimizer with a learning rate of $2 \times 10^{-4}$, employing linear warm-up for 16,000 steps followed by square root decay. We used a batch size of 128 and trained for 500,000 steps on freshly generated synthetic data, ensuring no repetition of examples. Training required approximately one day for the string manipulation domain and around five hours for the list manipulation domain, using 8 TPU v2 accelerators per model. (Shi et al., 2023b)

## B  PURELY INDUCTIVE PROGRAM SYNTHESIS

To understand where and when transductive guidance helps and when it hurts, we consider an induction-only ablation. This algorithm (Algorithm 2) follows the same execution-guided synthesis procedure as ExeDec but omits all transductive predictions, relying purely on inductive reasoning to construct programs from the task specification. In this way, the setup represents a purely inductive ablation of ExeDec without any form of guidance.

### B.1  ABLATION ALGORITHM

At each step $k$, the inductive model looks at the current state of the task and proposes a candidate subprogram, i.e., a program consisting of a single DSL primitive together with its arguments. The state in the first iteration is simply the original I/O specification $\mathcal{S} = \{(I_0, O_0), \ldots, (I_n, O_n)\}$, which pairs each input with its desired output. Once the inductive model generates a subprogram, it is executed on all inputs $\{I_i\}$, producing a set of outputs $\{\hat{O}_i\}$. The algorithm then checks whether these predicted outputs match the target outputs $\{O_i\}$. If the outputs already match, a program is found: the subprograms collected so far form a complete program that maps every input to its respective output. If the outputs do not match, the algorithm updates the state so that the next step can focus on the remaining gap between the current outputs and the target outputs. The way this update is done depends on the domain. In the list domain, the execution results $\{\hat{O}_i\}$ are treated as the new inputs while the original target outputs $\{O_i\}$ remain fixed, yielding the new state $\{(\hat{O}_i, O_i)\}$. In the string domain, the inputs $\{I_i\}$ stay unchanged, but the outputs are shortened by subtracting the part already produced by the program, resulting in the new state $\{(I_i, O_i - \hat{O}_i)\}$. After each state

---

**Algorithm 2** Ablation algorithm using no transductive guidance at all.
Note, $\{x_i\}$ is short for $[x_1, \ldots, x_n]$, where $n$ is the number of I/O pairs.

---

**Require:** Inductive step limit $K$
**Ensure:** full_program
  1: Initialize full_program $\leftarrow$ []
  2: current_state $\leftarrow \{(I_i, O_i)\}$
  3: **for** $k = 1$ to $K$ **do**                                          $\triangleright$ Inductive synthesis
  4:      subprogram $\leftarrow$ InductiveModel(current_state)
  5:      $\{\hat{O}_i\} \leftarrow$ Execute(subprogram, $\{I_i\}$)
  6:      **if** $\{\hat{O}_i\} = \{O_i\}$ **then**                          $\triangleright$ Found successful program
  7:         full_program $\leftarrow$ subprogram
  8:         **return** full_program
  9:      **if** domain = "list" **then**                         $\triangleright$ Domain-specific updates
10:         current_state $\leftarrow \{(\hat{O}_i, O_i)\}$
11:      **else if** domain = "string" **then**
12:         current_state $\leftarrow \{(I_i, O_i - \hat{O}_i)\}$
13:      full_program $\leftarrow$ subprogram
14: **return** full_program

---

update, the generated subprogram is appended to the sequence of previously generated subprograms. This way, the full program is built up incrementally, step by step. The process continues for at most $K$ steps, either until a correct program is found or until the step limit is reached.

**Model Training.** Training data is generated by randomly sampling programs from the DSL, with each program decomposed into a sequence of subprograms as described in Appendices A.1.2 and A.2.2. The synthesis model used to create the results displayed in the main paper in Section 3, is trained exactly like ExeDec's No-Subgoal Ablation Model: Given a *task* specification predict the subprogram that solves the first subtask. Thus, the model can learn to use DSL primitives and additionally sees training tasks that require more than one subprogram to be solved. Consequently, this model learns to decide which step to take first and thus reason about solving complex tasks. Contrary to their ablation, we evaluate all approaches on held-out test cases. Besides using the No-Subgoal Ablation model as a synthesis model, we also use ExeDec's Synthesizer Model in our ablation which is also trained to predict subprograms yet given a *subtask* specification. Consequently, this model only learns to use DSL primitives but is not trained on tasks that require more than one subprogram to be solved thus lacking any planning skills.

## B.2 ABLATION USING EXEDEC'S SYNTHESIZER MODEL

In this section, we report ablation results using the Synthesizer Model rather than the No-Subgoal Ablation Model. Unlike the No-Subgoal Ablation Model, the Synthesizer Model is trained to generate subprograms given *subtask* specifications. While it still mirrors the algorithmic workflow of ExeDec, it does not leverage permanent subtask decomposition for transductive guidance. Since the Synthesizer Model is trained only on subtasks, it learns how to use DSL primitives but does not acquire the ability to reason about complex, multi-step solution strategies, allowing us to examine the effect of permanent transductive guidance under these limitations.

**Transductive Guidance Can Hurt Performance.** As the first step in understanding the role of transductive guidance in program synthesis, we analyzed how often ExeDec invokes its guidance module, i.e., the number of subtask decompositions performed during synthesis. Figure 11 shows the number of ExeDec guidance calls in solved tasks compared to the ground-truth number of subtask decompositions across different domains and generalization categories. In the list domain, ExeDec

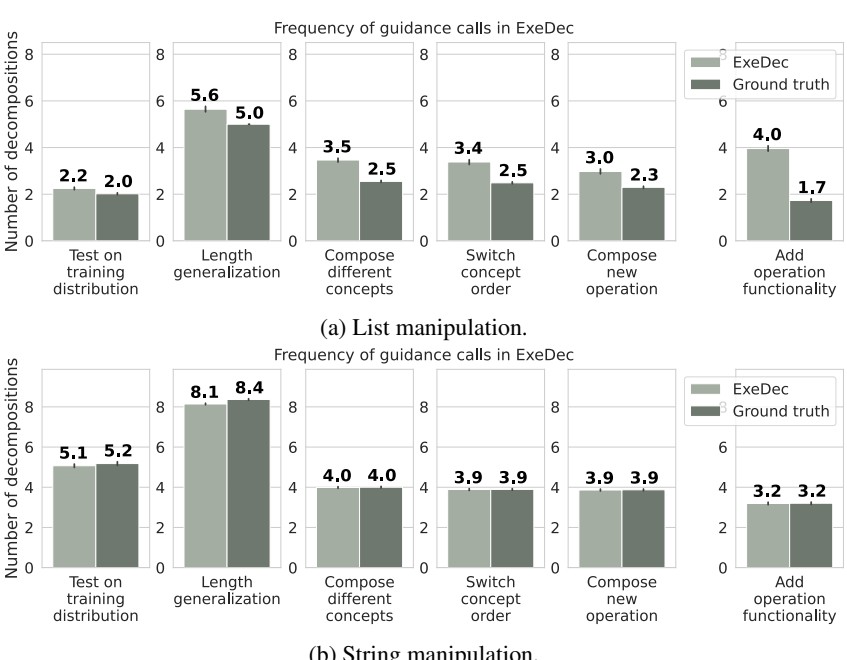

(a) List manipulation.

(b) String manipulation.

Figure 11: Number of calls to the transductive Decomposition model in ExeDec.

consistently uses more guidance calls than are strictly necessary. That is, the number of predicted decompositions exceeds the ground-truth number across all categories. This overuse becomes especially pronounced in generalization categories, where ExeDec invokes more than twice the necessary number of decompositions. In contrast, in the string domain, ExeDec closely matches the ground-truth number of decompositions. This is expected because, as discussed in Appendix A.1, only a single or very few decomposition sequences are valid for each task in this domain. The input remains constant across subprograms, and tasks are generated by appending subprograms, which enforces a largely fixed decomposition pattern. Taken together, these findings demonstrate that transductive guidance in the form of subtask decompositions exhibits domain-specific utility: in domains where subtasks are largely independent, guidance calls provide an efficient means to solve tasks, whereas in domains with tightly nested decomposition patterns, their efficiency is ambiguous.

This observation prompted a more detailed investigation into the role of transductive guidance in ExeDec. To this end, we developed an ablation algorithm that mirrors ExeDec's procedure but omits calls to the decomposition model, effectively removing transductive guidance. Figure 12 shows performance across both domains using the same synthesis model as in ExeDec. In the string domain, ExeDec achieves an average generalization accuracy of $73.92\% \pm 12.36\%$, while the ablation performs substantially worse at $6.84\% \pm 12.29\%$. This large difference arises because tasks in the string manipulation domain typically admit only one or a small number of valid subtask sequences. Solving these tasks requires long-term planning and coordination across multiple steps. Since the Synthesizer Model is trained solely on single-step tasks, it lacks this planning capability, explaining its poor performance. In the list domain, the ablation performs similarly to ExeDec ($12.01\% \pm 6.52\%$ vs. $12.19\% \pm 4.97\%$). At the category level, the ablation even outperforms ExeDec in Compose New Operation, but performs worse in Length Generalization. Solutions in the list domain typically allow multiple valid decompositions, which means that transductive guidance can sometimes overwrite otherwise correct inductive reasoning, reducing the benefit of permanent guidance. Conversely, in domains or categories where a single semantic decomposition exists or

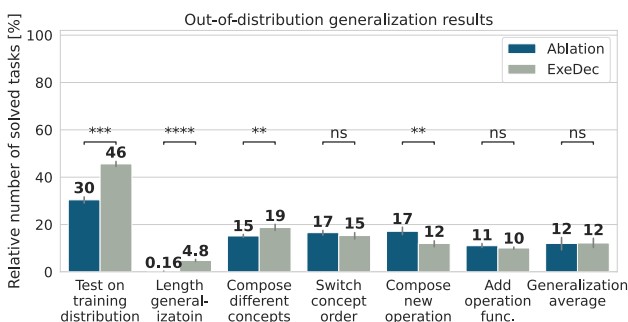

(a) List manipulation.

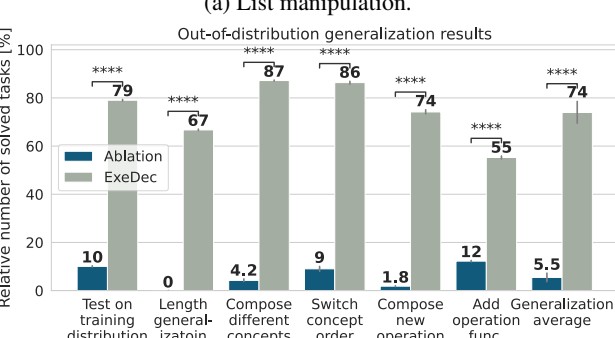

(b) String manipulation.

Figure 12: Performance of ExeDec compared to the ablation using the Synthesizer Model. While ExeDec clearly outperforms the ablation on the string domain, it performs similar to the ablation in the list domain. This shows the ambiguous benefits of permanent transductive guidance.

long-term planning is required (e.g., string tasks or length generalization), transductive guidance remains essential.

## C    TRANSDUCTIVELY INFORMED INDUCTIVE PROGRAM SYNTHESIS

Our ablation study of ExeDec reveals a key finding: transductive guidance is important for program synthesis but can also be hindering at times. Specifically, permanent transductive guidance forces a trajectory based on transductive skills onto the inductive synthesizer, pushing it to solve different tasks than those it would naturally address without guidance. This permanent guidance can help but also overrule inductive reasoning, thus misleading the inductive synthesizer for tasks that require inductive reasoning to be solved.

### C.1    TIIPS ALGORITHM

TIIPS exploits this learning by applying incremental transductive guidance *only when the inductive approach fails* to yield a successful program. Additionally, TIIPS *incrementally increases* the degree of transductive guidance with each failed attempt, rather than employing permanent transductive guidance throughout the entire synthesis process. This selective approach allows us to harness the benefits of transductive guidance while avoiding its potential drawbacks.

To illustrate this process, consider the analogy of a data scientist collaborating with a finance expert. The finance expert tasks the data scientist to implement a data processing pipeline that aggregates multiple features into a recommendation to buy a stock or not. The data scientist initially attempts to implement this entire pipeline independently, breaking it down into subtasks: data selection, data filtering, and feature aggregation. However, due to missing domain knowledge, the data scientist fails at multiple steps: first, not knowing which financial features are relevant for stock recommendations, and second, lacking knowledge about the specific filtering criteria beyond removing missing data. In the next iteration, the data scientist asks the finance expert to provide examples of what the output should look like after the data selection step, learning which features (such as price-to-earnings ratio, debt-to-equity ratio, and moving averages) are crucial for stock analysis. Using this guidance, the data scientist successfully implements the first step. However, the filtering step still fails because the scientist only removes missing data without applying domain-specific filters. In the subsequent iteration, the data scientist requests guidance for the filtering step, asking the finance expert to show examples of what the filtered output should look like. With this additional guidance, the data scientist learns to filter out stocks with extreme volatility or insufficient trading volume. The scientist then proceeds to implement the remaining aggregation step without additional guidance, noticing that the data is not normally distributed and consequently normalizing the data before aggregating it into a recommendation. This demonstrates how selective guidance can enable expert knowledge where needed while preserving the data scientist's analytical reasoning for other steps.

Translating this analogy to technical terms, the TIIPS algorithm operates through nested loops as shown in the pseudocode in Algorithm 3. The inner loop corresponds to purely inductive synthesis attempting to solve the entire task, while the outer loop implements the incremental guidance mechanism. In the first outer iteration ($t \leq k$), the algorithm attempts purely inductive synthesis with the full I/O specification $\mathcal{S} = \{(I_0, O_0), \ldots, (I_n, O_n)\}$. The inner loop iterates for up to $K$ steps ensuring the final program does not exceed maximum length $K$. The inner loop operates identically to the algorithm from our ablation study: at each step $k$, the inductive model generates a subprogram based on the current state, executes it on all inputs to produce outputs $\{\hat{O}_i\}$, and checks if the program matches the task specifications. If the task specifications are not met, the algorithm updates the state and appends the generated subprogram to the growing full program. The state update procedure differs by domain to reflect how outputs are generated: for the list manipulation domain, the execution results become new inputs while target outputs remain fixed, whereas for the string manipulation domain, the execution result is subtracted from the target output with inputs unchanged. Regardless of domain, each generated subprogram represents a computational step that brings the current state closer to the target, and these subprograms are sequentially concatenated to form the complete solution program. If *and only if* the inner loop fails to find a solution within the remaining steps, the outer loop activates transductive guidance. The transductive model predicts new output samples $\{\bar{O}_i\}$ based on the stored task specification, creating a refined subtask $\{(I_i, \bar{O}_i)\}$ that focuses the inductive model on the next subgoal. If the inductive model does not find a solution in the remaining

---

**Algorithm 3** TIIPS algorithm using sparse transductive guidance when needed. Note, $\{x_i\}$ is short for $[x_1, \ldots, x_n]$, where $n$ is the number of I/O pairs.

---

**Require:** Transductive step limit $T$, Inductive step limit $K$
**Ensure:** full_program
 1: Initialize full_program $\leftarrow []$
 2: current_state $\leftarrow \{(I_i, O_i)\}$
 3: **for** $t = 1$ to $T$ **do** ▷ Outer loop: transductive retries
 4:     current_state $\leftarrow \{(I_i, O_i)\}$ ▷ Reset state to initial task specification
 5:     **for** $k = 1$ to $K$ **do** ▷ Inner loop: inductive synthesis
 6:         **if** $k < t$ **then**
 7:             $\{\bar{O}_i\} \leftarrow$ TransductiveModel(current_state) ▷ Use transductive guidance
 8:         **else**
 9:             $\{\bar{O}_i\} \leftarrow \{O_i\}$ ▷ No transductive guidance
10:         current_state $\leftarrow \{(I_i, \bar{O}_i)\}$
11:         subprogram $\leftarrow$ InductiveModel(current_state) ▷ Inductive program generation
12:         $\{\hat{O}_i\} \leftarrow$ Execute(subprogram, $\{I_i\}$)
13:         **if** $\{\hat{O}_i\} = \{O_i\}$ **then** ▷ Found successful program
14:             full_program $\leftarrow$ subprogram
15:             **return** full_program
16:         **if** domain = "list" **then** ▷ Domain-specific updates
17:             current_state $\leftarrow \{(\hat{O}_i, O_i)\}$
18:         **else if** domain = "string" **then**
19:             current_state $\leftarrow \{(I_i, O_i - \hat{O}_i)\}$
20:     full_program $\leftarrow$ subprogram
21: **return** full_program

---

$K - t$ steps, the process repeats for up to $T$ outer loop iterations, with each iteration providing guidance for $t$ steps while allowing the inductive model to handle the remaining steps independently. The algorithm terminates either when a complete solution is found or when the maximum number of transductive attempts is reached.

Unlike TIIPS, ExeDec integrates transductive predictions rigidly at every step, forcing the inductive model to follow a fixed transductive trajectory. While this can provide strong guidance, it often overconstrains the search: even small prediction errors may block the inductive model from discovering a correct subprogram, despite one existing. In doing so, ExeDec also discards the potential of pure inductive reasoning. TIIPS, in contrast, can be seen as a step-wise ensemble of ExeDec and the induction-only ablation. It starts from induction alone and only introduces guidance when inductive synthesis fails, ranging from no guidance at all (equivalent to the ablation) to applying guidance at every step (equivalent to ExeDec), with the flexibility to activate guidance for any number of steps in between. This approach allows TIIPS to gradually increase the level of guidance across iterations, but never applies it more strictly than required. In this way, it preserves the strengths of inductive reasoning while still offering transductive fallback when needed, avoiding the rigid overconstraint of ExeDec.

## C.2 Illustrative Comparison of TIIPS and ExeDec Workflow

Figure 13 illustrates how TIIPS solves a list manipulation task that neither ExeDec nor the induction-only ablation could complete. In this example, TIIPS exploits transductive guidance in the generation of the first subprogram, using the guidance prediction to decompose the task at the initial step. The remaining steps are then completed without further transductive guidance, relying solely on the outputs of previous subprograms and standard induction. The ground truth program $x_0 =$ INPUT | $x_1 =$ Map (*3) $x_0$ | $x_2 =$ Scanl1 (max) $x_1$ | $x_3 =$ ZipWith (-)$x_1 x_2$ differs from the program found by TIIPS. Notably, the transductive guidance for the first step deviates from the ground truth. Nevertheless, TIIPS is able to leverage this guidance to solve the task correctly, indicating that the initial guidance does not mislead later steps that rely on inductive reasoning. In contrast, ExeDec also depends on transductive guidance for subsequent steps but fails

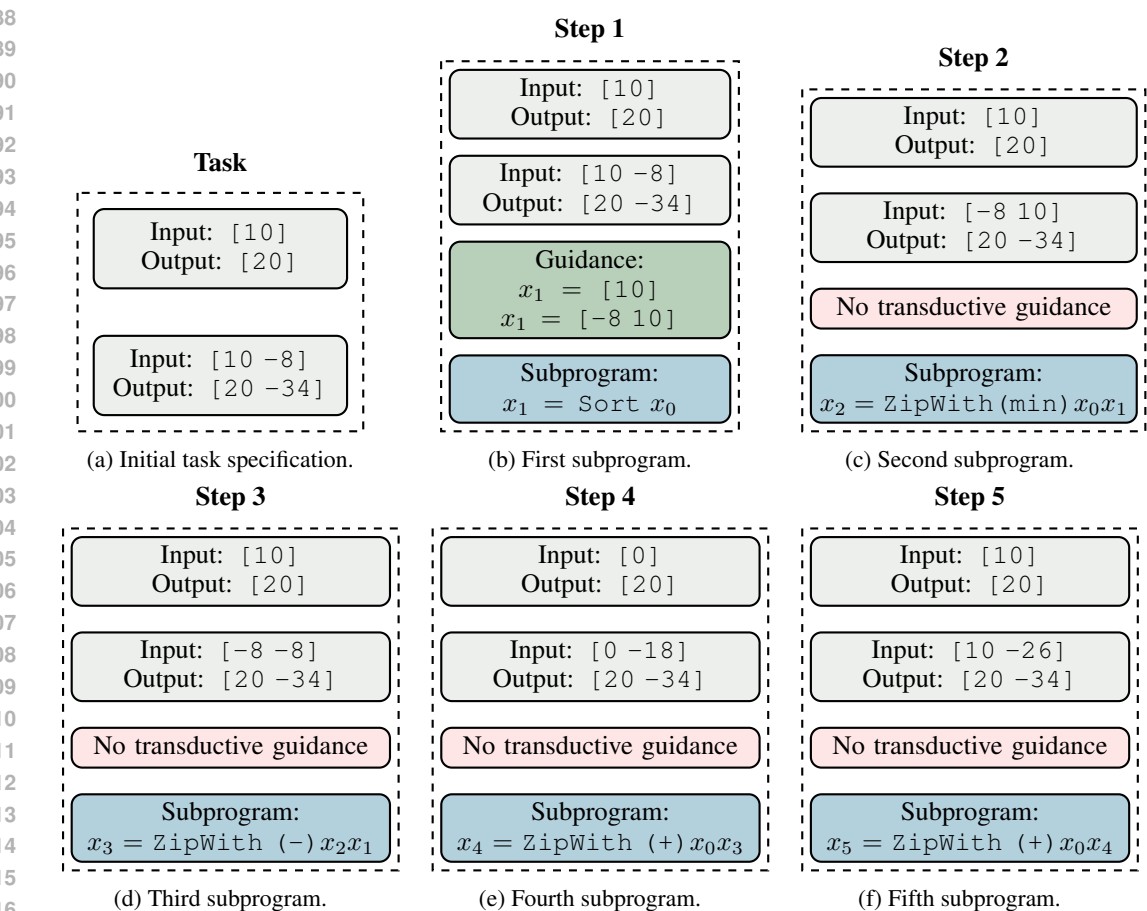

Figure 13: Real task taken from the list manipulation domain. The task was solved by TIIPS exclusively. The workflow illustrates how TIIPS exploited transductive guidance to generate the first subprogram after failing to generate a solution without any guidance. The remaining steps were solved without further transductive guidance. This example underscores the significance of strategic, light-touch guidance that supports inductive reasoning without being overly restrictive. Note, $x_0$ is the initial task input.

to recover from an alternative first-step guidance, preventing it from reaching a correct solution. This workflow demonstrates that TIIPS does not require persistent transductive guidance for every subtask; instead, a sparse, targeted use of guidance is sufficient to 'kickstart' the decomposition. In contrast, ExeDec fails to solve this task because the guidance predictions for one or more subtasks after the first step are misleading, which leads to incorrect guidance and prevents task completion. Thus, the figure highlights the utility of sparse transductive guidance as a catalyst enabling the synthesis process to start correctly, rather than as a crutch.

## C.3 TIIPS ANALYSIS

Our results in the main paper show the effectiveness of guidance in TIIPS and the varying importance depending on both the domain and the type of task, highlighting the flexible role of transductive support. TIIPS matches or outperforms ExeDec in both domains. In the string domain, performance is nearly identical, while in the list domain, TIIPS clearly outperforms ExeDec, solving substantially more tasks. The $p$ and $t$ values of the paired t-tests are displayed in Table 1 and Table 2. Analysis of task overlaps shows that TIIPS solves many tasks exclusively, particularly in the list domain, whereas the differences in the string domain are minimal. These findings indicate that TIIPS effectively balances transductive guidance with inductive reasoning, leveraging full guidance when

| Category | ExeDec vs No-Subgoal Ablation | ExeDec vs Synthesizer Ablation | ExeDec vs TIIPS |
|---|---|---|---|
| Test on training distribution | $p = 9.73\times10^{-1}, t = -0.04$ | $p = 3.48\times10^{-4}, t = -11.3$ | $p = 7.52\times10^{-2}, t = -2.39$ |
| Length Generalization | $p = 1.38\times10^{-2}, t = 4.19$ | $p = 2.28\times10^{-5}, t = -22.6$ | $p = 3.69\times10^{-2}, t = -3.08$ |
| Compose different concepts | $p = 4.52\times10^{-3}, t = -5.76$ | $p = 2.80\times10^{-3}, t = -6.56$ | $p = 1.32\times10^{-4}, t = -14.5$ |
| Switch Concept Order | $p = 1.30\times10^{-1}, t = -1.90$ | $p = 1.38\times10^{-1}, t = 1.85$ | $p = 1.02\times10^{-5}, t = -27.7$ |
| Compose new operation | $p = 7.25\times10^{-3}, t = -5.05$ | $p = 1.35\times10^{-3}, t = 7.95$ | $p = 3.62\times10^{-4}, t = -11.2$ |
| Add operation functionality | $p = 6.97\times10^{-1}, t = -0.42$ | $p = 2.14\times10^{-1}, t = 1.48$ | $p = 1.53\times10^{-4}, t = -14.0$ |
| Generalization average | $p = 4.17\times10^{-3}, t = -3.17$ | $p = 1.24\times10^{-9}, t = -9.54$ | $p = 8.15\times10^{-1}, t = -0.24$ |

Table 1: Paired t-tests on the list domain. Each cell shows $p$-value and $t$-statistic.

| Category | ExeDec vs No-Subgoal Ablation | ExeDec vs Synthesizer Ablation | ExeDec vs TIIPS |
|---|---|---|---|
| Test on training distribution | $p = 1.40\times10^{-1}, t = -1.84$ | $p = 5.28\times10^{-10}, t = -326$ | $p = 2.42\times10^{-1}, t = 1.37$ |
| Length Generalization | $p = 1.11\times10^{-1}, t = -2.04$ | $p = 3.71\times10^{-10}, t = -356$ | $p = 4.62\times10^{-4}, t = 10.5$ |
| Compose different concepts | $p = 1.06\times10^{-3}, t = -8.48$ | $p = 6.68\times10^{-10}, t = -308$ | $p = 5.40\times10^{-3}, t = 5.48$ |
| Switch Concept Order | $p = 7.60\times10^{-1}, t = 0.33$ | $p = 9.04\times10^{-9}, t = -161$ | $p = 1.67\times10^{-4}, t = 13.7$ |
| Compose new operation | $p = 9.91\times10^{-5}, t = -15.6$ | $p = 1.77\times10^{-8}, t = -136$ | $p = 7.76\times10^{-3}, t = 4.95$ |
| Add operation functionality | $p = 1.79\times10^{-1}, t = 1.63$ | $p = 7.04\times10^{-9}, t = -171$ | $p = 7.51\times10^{-3}, t = 5.00$ |
| Generalization average | $p = 1.95\times10^{-3}, t = -3.45$ | $p = 2.17\times10^{-18}, t = -24.25$ | $p = 2.2\times10^{-10}, t = 1.04$ |

Table 2: Paired t-tests on the string domain. Each cell shows $p$-value and $t$-statistic.

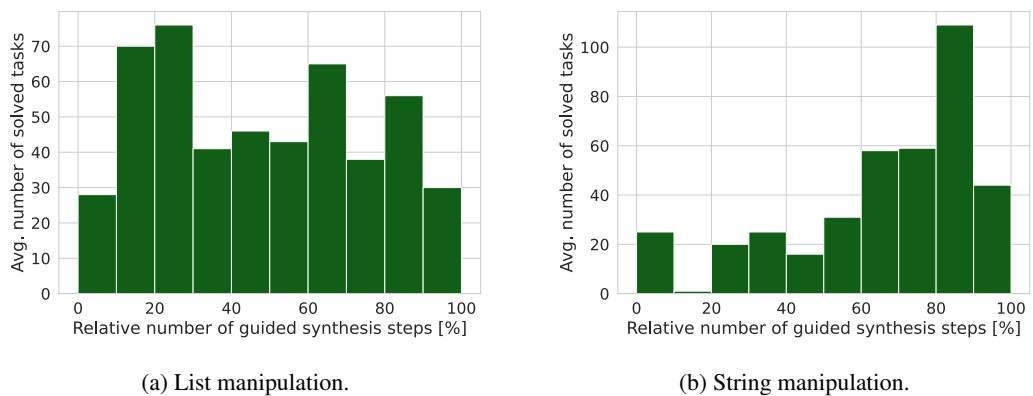

(a) List manipulation.

(b) String manipulation.

Figure 14: Distribution of the degree of guidance of tasks solved by TIIPS. The x-axis shows the number of guided steps relative to the number of steps the discovered solution consists of.

beneficial but retaining flexibility to use partial or no guidance where permanent guidance is less helpful.

This finding is further supported by the results displayed in Figure 14. In the string manipulation domain, most problems are solved using full guidance, reflecting the benefit of guiding every step. Purely inductive reasoning contributes very little, underscoring the difficulty of solving these tasks without guidance. Overall, the distribution of guidance degree shows a right skew, suggesting that guidance plays a more significant role in this domain. However, as there is also a substantial number of tasks solved using partial or no guidance, it also demonstrates that selectively guiding steps can provide substantial benefits, even when full guidance is not applied. By contrast, in the list manipulation domain, more tasks can be solved without guidance. Full guidance has little impact, consistent with the findings from our ablation study, indicating that permanent transductive guidance adds little value. Partial guidance, however, is employed across a notable number of tasks and is more evenly distributed across categories than in the string domain. Moreover, the distribution is more uniformly than in the string domain. This suggests that a selective approach to guidance can complement inductive synthesis effectively, even when baseline performance is already strong. Taken together, these results show that the degree and timing of guidance are both domain and task-dependent. Full guidance is crucial for string tasks, while list tasks often benefit from minimal or partial guidance. Partial guidance emerges as a flexible strategy, striking a balance between transductive hints and the

natural problem-solving ability of inductive synthesis, particularly in intermediate cases where full guidance is unnecessary but no guidance alone may be insufficient.

Beyond quantitative performance, we examine the robustness of generated solutions in the both domains using density plots. Each solved task is represented by a point in a two-dimensional space, with intent match on the x-axis and syntactical overlap on the y-axis. Intent match measures semantic alignment between predicted and ground-truth subtasks by comparing execution results: Formally, let $T = \{t_1, \ldots, t_n\}$ be the set of ground-truth subtasks for a given task, and $\hat{T} = \{\hat{t}_1, \ldots, \hat{t}_m\}$ be the predicted subtasks (or the execution results of predicted subprograms in unguided synthesis). Intent match is defined as

$$I = \frac{|T \cap \hat{T}|}{|T|} \cdot 100\%,$$

where $|T \cap \hat{T}|$ is the number of correctly predicted subtasks (or execution-matching outputs). Syntactical overlap captures the structural similarity of the generated programs to the ground-truth solutions. Formally: Let $P = \{p_1, \ldots, p_n\}$ be the set of ground-truth subprograms, and $\hat{P} = \{\hat{p}_1, \ldots, \hat{p}_m\}$ be the predicted subprograms. The *syntactical overlap* is defined as

$$S = \frac{\left|\{p_i \in P \mid \exists \hat{p}_j \in \hat{P} : p_i = \hat{p}_j\}\right|}{|P|} \cdot 100\%.$$

For example, if a four-step task has three correctly predicted subtasks and two syntactically aligned subprograms, the corresponding point would lie at $(75\%, 50\%)$ in the plot.

In the list domain, TIIPS produces solutions that cluster more tightly in the top-right region than those of ExeDec. This suggests that TIIPS not only solves a larger fraction of tasks but also produces solutions that better preserve both semantic intent and syntactic form. A similar tendency can be observed in the string domain (Figure 15), though the effect is less pronounced than in the list domain. In general, the density distributions of solved tasks are broadly similar for TIIPS and ExeDec, indicating that both approaches generally produce semantically and syntactically accurate solutions. However, the distribution of TIIPS solutions is slightly skewed to the right along the

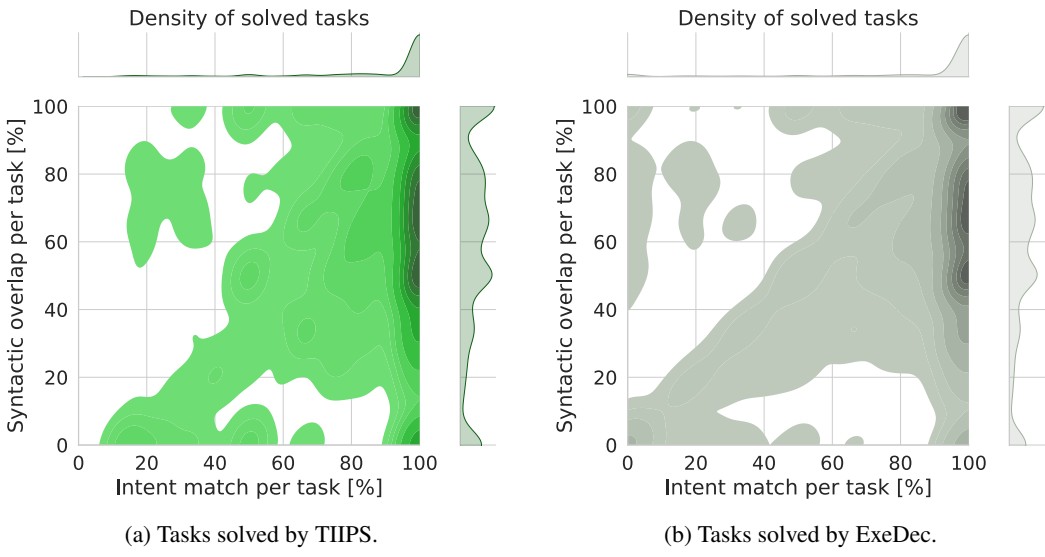

(a) Tasks solved by TIIPS.  (b) Tasks solved by ExeDec.

Figure 15: Tasks solved by TIIPS and ExeDec in the string manipulation domain, grouped by intent match (x-axis) and syntactical overlap (y-axis). Intent match measures the overlap between predicted/executed and ground-truth subtask outputs, while syntactical overlap reflects the structural similarity of predicted programs to ground-truth solutions. As a result, correctly solved tasks, both semantically and syntactically, concentrate in the top-right region. Results of both approaches are averaged across over 21,000 solved tasks spanning all compositional generalization categories and seeds.

x-axis, showing a tendency toward higher intent match. This suggests that while both methods generate solutions of comparable syntactic quality, TIIPS more consistently captures the semantic intent of subtasks. Overall, these results indicate that TIIPS not only maintains performance parity with ExeDec in the string domain but also produces solutions that slightly better preserve semantic alignment with the ground-truth programs.

We quantified how much more TIIPS occupies the top-right region of the density plots by dividing each plot into four quadrants. For instance, the top-right quadrant contains tasks with both intent match and syntactical overlap above 50%. In the list manipulation domain, more than twice as many solved tasks lie in the right quadrants for TIIPS compared to ExeDec, indicating that TIIPS predicts subtasks with intent match above 50% for a substantially larger portion of tasks. The top-right quadrant, which captures tasks with both high syntactic and semantic alignment to the ground truth, also contains approximately twice as many tasks for TIIPS as for ExeDec. This demonstrates that TIIPS produces solutions that are both semantically correct and syntactically closer to the ground truth. In the string manipulation domain, TIIPS shows slightly fewer tasks in the top-right quadrant compared to ExeDec. However, slightly more tasks are located in the right half of the plot (a factor of 1.02), meaning that TIIPS solves tasks with subtasks more aligned with the ground truth, even if it sometimes produces alternative implementations of the subprograms.

Overall, these results indicate that TIIPS generates clearly more robust programs in the list domain, aligning closer to the ground truth in both syntax and semantics. In the string domain, TIIPS slightly improves the robustness of programs generated by ExeDec, but this improvement is marginal, as programs produced by both approaches already align with the ground truth's syntax and semantics in almost all cases.

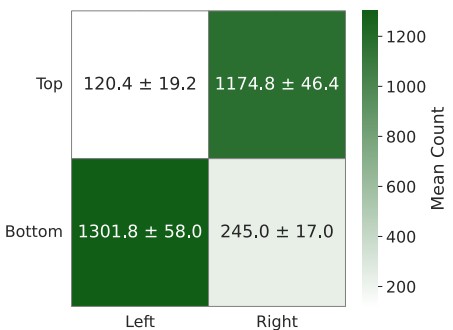
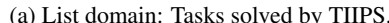

(a) List domain: Tasks solved by TIIPS.

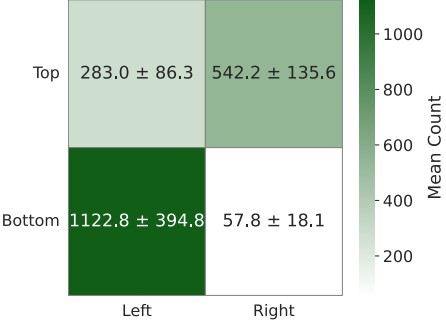
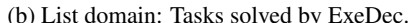

(b) List domain: Tasks solved by ExeDec.

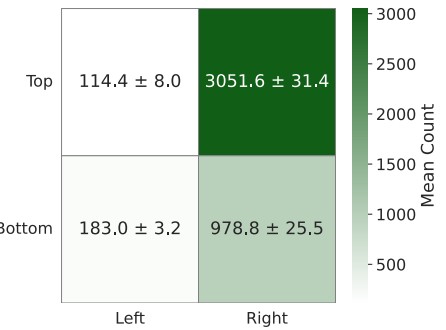

(c) String domain: Tasks solved by TIIPS.

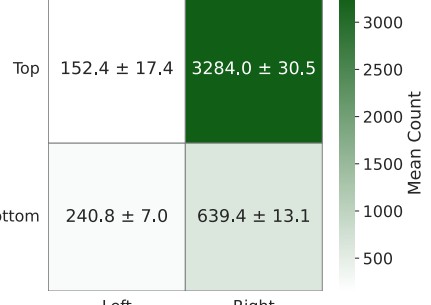

(d) String domain: Tasks solved by ExeDec.

Figure 16: Tasks solved by TIIPS and ExeDec in the string manipulation domain, grouped by intent match (x-axis) and syntactical overlap (y-axis). Tasks solved by TIIPS are clearly more often in the top right quadrant, i.e., closely align with the ground truth in syntax and semantics. Tasks solved by TIIPS are slightly more often in the right hemisphere, i.e., closely align with the ground truth in semantics.

## C.4 ADDITIONAL ANALYSIS & RESULTS

### C.4.1 COMPUTATIONAL RESOURCES

To assess and compare the computational resources required by TIIPS, ExeDec, and the two ablation models, we rerun the leave-one-out evaluation for all four approaches. For clarity, we refer to our inductive ablation which uses the Synthesizer Model trained on subtask specifications as the Ablation. The ablation introduced in the ExeDec paper is referred to as the No-Subgoal Ablation. These two ablations probe different questions. The No-Subgoal Ablation evaluates whether explicit task decomposition via the Decomposition Model provides an advantage over training the Synthesizer Model to implicitly learn decomposition from full task specifications. It therefore measures the effect of explicit versus implicit decomposition and requires the model to learn task-level reasoning. In contrast, our Ablation assesses the overall impact of using the Decomposition Model at all. The Synthesizer Model in this ablation is trained only on subtask specifications, and thus learns to reason at the level of individual subtasks. This ablation isolates the effect of removing decomposition entirely, rather than the effect of removing explicit decomposition alone.

Except for evaluating 100 instead of 1,000 test tasks per category, we follow the same evaluation setup as in the main experiments. To compare computational requirements across approaches fairly, we consider only tasks that all four methods solved across all five random seeds. This filtering yields 906 tasks in the string domain and 656 tasks in the list domain. We report three efficiency metrics that reflect complementary aspects of computational cost. Elapsed time measures the wall-clock inference time until a correct program is found. The number of evaluated programs counts how many candidate programs are generated and executed before discovering a solution. Finally, model calls capture how often the synthesis model and, where applicable, the guidance model are invoked. This metric is particularly relevant because model queries can be substantially more expensive than program execution.

Figure 17 shows the inference time across all six categories. Across most categories in the list domain, TIIPS solves tasks noticeably faster than ExeDec. This advantage is most pronounced in the out-of-distribution settings, where TIIPS benefits from starting inductively and introducing guidance only when necessary. The only exception is Length Generalization and the in-distribution test setting, where TIIPS requires slightly more time. This is expected: these categories demand longer planning horizons, and permanent guidance offers a direct advantage in such cases. Both ablation models are the most efficient overall, as they do not rely on guidance and therefore evaluate fewer guided trajectories. A similar pattern emerges in the string domain: the two ablations again achieve the lowest inference times. In contrast to the list domain, however, TIIPS is slower than ExeDec. On average, TIIPS requires roughly twice the inference time to solve a task. This difference is consistent with the characteristics of the domain and our other results. As Figure 14b shows, guidance is critical for achieving high generalization in string manipulation, and Figure 11b demonstrates that the guidance model produces highly accurate predictions in this domain. TIIPS therefore spends additional time exploring unguided or partially guided trajectories before converging on the fully guided strategy that ExeDec applies from the start.

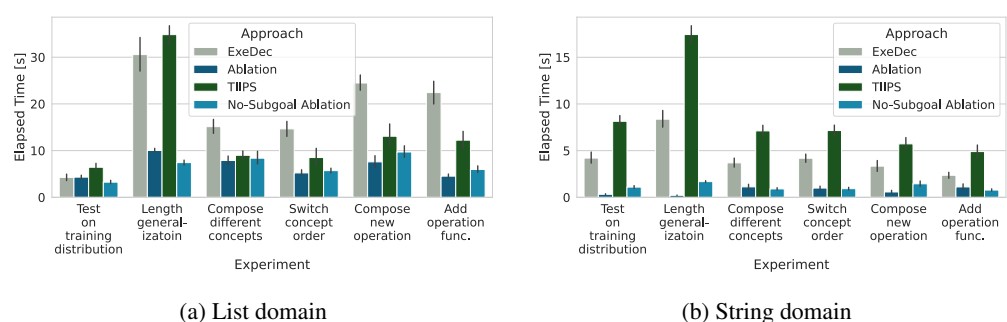

(a) List domain
(b) String domain

Figure 17: Inference time required by either approach to discover a solution. Both ablation model require the least time. TIIPS is more efficient in the list domain but clearly less efficient in the string domain.

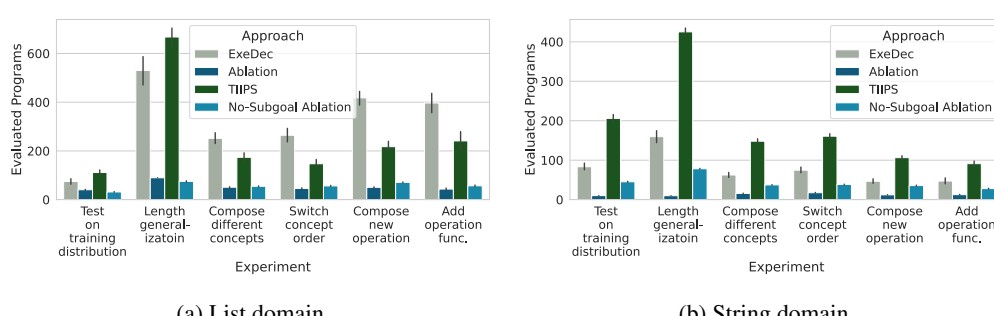

(a) List domain

(b) String domain

Figure 18: Number of evaluated programs across approaches. While TIIPS evaluates fewer programs than ExeDec in the list domain, it requires roughly twice as many evaluated programs as ExeDec in the string domain, reflecting the stronger role of guidance and the higher accuracy of guidance predictions in this domain.

The number of evaluated programs mirrors the inference-time analysis in both domains (Figure 18). In the list domain, TIIPS evaluates fewer programs than ExeDec before finding a solution in all out-of-distribution categories except Length Generalization. This confirms that TIIPS benefits from beginning inductively and relying on guidance only when it becomes necessary, which reduces the number of trajectories explored. The same pattern as in inference time also appears in the string domain. TIIPS evaluates roughly twice as many programs as ExeDec to discover a correct solution. This is consistent with the stronger role of guidance in the string domain, where ExeDec's fully guided search is more efficient, and TIIPS spends additional effort exploring partially guided or unguided program candidates.

The number of model calls (Figure 19) paints the same picture as the inference time and number of evaluated programs: TIIPS requires fewer model calls than ExeDec in the list domain. The main exception is Length Generalization, where nearly every step demands guidance, resulting in substantially more calls. TIIPS also issues slightly more calls in the Compose-New-Operation category, though the difference is minor. As with the other two metrics, both ablation models make the fewest model calls overall, since they operate entirely without guidance. The same holds true for the string domain: TIIPS requires more than twice as many model calls as ExeDec to solve a task. This again reflects the importance of guidance in the string domain and the high accuracy of the guidance model, which makes ExeDec's fully guided search more efficient in this domain.

In conclusion, TIIPS solves tasks more efficiently than ExeDec when purely inductive approaches have an edge. In settings where guidance is essential, TIIPS ultimately reverts to ExeDec's fully guided behavior. However, its incremental increase of guided steps leads to additional, non-expedient synthesizer calls. These results highlight the importance of developing methods that can reliably decide *when* to apply guidance and *when* to rely on inductive reasoning alone.

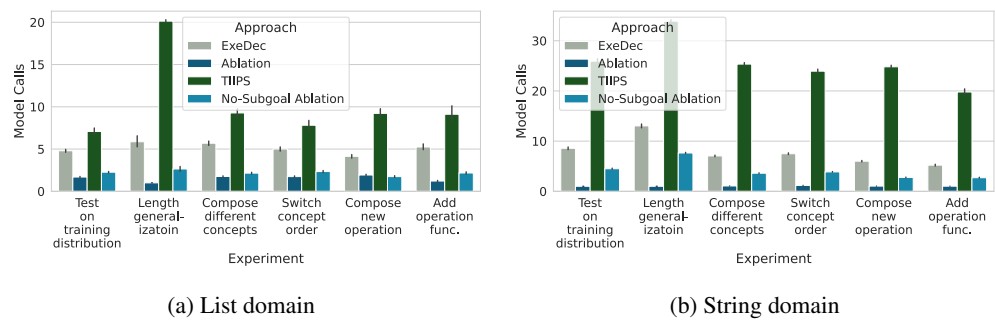

(a) List domain

(b) String domain

Figure 19: Number of model calls across approaches. In the string domain, TIIPS generally requires fewer model calls than ExeDec. In the string domain, TIIPS requires more than twice as many model calls as ExeDec–mirroring the patterns observed for inference time and evaluated programs.

## C.5 Additional String Manipulation Benchmarks

We evaluated ExeDec and TIIPS on two additional string manipulation benchmarks, SyGuS (Shi et al., 2022) and PROSE (Microsoft, 2022), to assess the generality of our approach. Both benchmarks consist of human-created or human-evaluated tasks, and PROSE tasks are among the most challenging string editing benchmarks, while SyGuS tasks are easier and comparable in difficulty to the Robustfill tasks used in our main evaluation (Li & Ellis, 2024; Cambronero et al., 2023).

For evaluation, we used models trained on the Test-on-Training distribution of the Robustfill domain. For SyGuS, we extracted tasks from the 2019 PBE SLIA Track and the 2018 PBE Strings Track in which both the inputs and outputs were single strings. We removed tasks that had more than one input string and duplicate tasks that differed only in the input string, which resulted in 74 unique tasks (Odena et al., 2020). For PROSE, we extracted tasks from the PROSE Text Transformation Track using the same criteria as for SyGuS. We retained only tasks that transform the input string without adding new strings, yielding 162 tasks.

Figure 20 shows the performance of TIIPS, ExeDec, and the induction only ablation on both benchmarks. On SyGuS, ExeDec achieves an accuracy of 53.51% ± 3.39%, while TIIPS achieves 58.65% ± 1.21%; the baseline achieves 49.19% ± 2.26%. On PROSE, ExeDec achieves 43.09% ± 1.65%, TIIPS achieves 47.28% ± 1.93%, and the baseline achieves 39.16% ± 0.94%. These results indicate that TIIPS consistently outperforms ExeDec on these string manipulation benchmarks that were not used for training. These results are surprising because ExeDec and TIIPS perform on par in the Robustfill domain, which was used for training. On the SyGUS and PROSE benchmarks, however, TIIPS consistently outperforms ExeDec. This can be explained by the decomposition model being less certain on these domains. On tasks that are harder to solve or where the decomposition model is uncertain, TIIPS leverages the full power of the inductive synthesis model, exploring solutions beyond the transductive trajectories prescribed by ExeDec. This allows TIIPS to achieve higher accuracy on challenging string manipulation tasks and reinforces the findings observed in the list manipulation domain. We conducted paired t-tests to evaluate statistical significance 3.

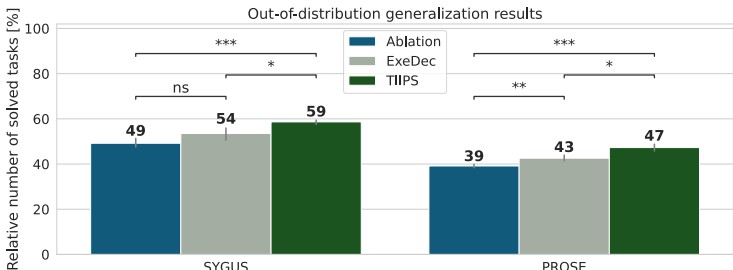

Figure 20: Number of solved tasks on the SyGuS and PROSE benchmarks. TIIPS outperforms ExeDec and the Ablation.

| Benchmark | ExeDec vs TIIPS | Baseline vs TIIPS | Baseline vs ExeDec |
|---|---|---|---|
| PROSE | $p = 1.36 \times 10^{-2}, t = -4.209$ | $p = 6.25 \times 10^{-4}, t = -9.731$ | $p = 8.64 \times 10^{-3}, t = -4.802$ |
| SYGUS | $p = 1.15 \times 10^{-2}, t = -4.417$ | $p = 3.79 \times 10^{-4}, t = -11.07$ | $p = 5.06 \times 10^{-2}, t = -2.764$ |

Table 3: Paired t-tests on the SyGUS and PROSE string benchmarks. Each cell shows the $p$-value and $t$-statistic.

