# OpenReview forum: "To Guide or Not to Guide: Sparse Transductive Guidance in Program Synthesis"
_ICLR.cc/2026/Conference — Submitted to ICLR 2026_

### Official Review · Reviewer_GjAY · 2025-10-27

**Soundness:** 3
**Presentation:** 3
**Contribution:** 2
**Rating:** 6
**Confidence:** 4

**Summary:**

The authors (1) show that transductive supervision of a certain kind of previously-proposed program synthesis system can hurt their performance and (2) propose an alternative framework that addresses this limitation while still leveraging the transductive and inductive modules of the prior work.

**Strengths:**

- The paper identifies precise limitations of prior work that motivate the reported investigations. In particular, the examination of the performance of transductive and inductive systems are valuable and well-designed.
- The methodology (both the experiments and the proposed framework) is well-motivated and supported with an ablation study of prior work.
- Results are communicated clearly.
- The proposed system is simultaneously conceptually simple and effective (relative to the prior work on which it is based).
- The work is clearly positioned with respect to prior work.
- The manuscript details amount of compute necessary to run the reported experiments

**Weaknesses:**

- The description of the proposed framework in the manuscript is hard to understand (section 4.1). I had to refer to the pseudocode in the appendix to understand it. I suggest revisiting this description and moving the pseudocode to the main text (the pseudocode is much easier to understand than both the natural language description and Figure 3).
- The proposed method and the significance of the results are limited by the substantial reliance of the synthesis algorithm on domain-specific structure (namely, the additive nature of strings and lists).
- The work is farily incremental in nature.

Minor writing issues:
- Section 2.2 should mention that the output is produced by computing prefixes (otherwise the example requires familiarity with ExeDec).

Minor typesetting weaknesses:
- Spacing between paragraphs is very small and makes the manuscript difficult to read.
- Quotation marks are used incorrectly (” is used to open quotations, instead of “)

**Questions:**

- What happens when the inductive model fails to solve the sub-task given by the transductive model? Line 12 of Alg 3.
- In line 349: "Tasks that TIIPS fails to solve but ExeDec succeeds on can be attributed to the last step not being guided: TIIPS may find a program that solves all training I/O pairs except the test pair, producing a false-positive program.".
	- Is this not the case for ExeDec too?
- To what extent do you think the results reported here can help the community design better synthesis systems for other domains?
- It seems the results correspond to a single "run" of each system (aggregated over a set of problems). However, there is some reliance on random numbers (e.g., ExeDec gets multiple attempts). Standard practice is to perform the experiments multiple times and aggregate to avoid spurious results and quantify variance. This would further strengthen the experiments.

---

> ### Author Response · Authors · 2025-11-18
> **Response 1/2**
>
> We thank the reviewer for insightful observations and address them below.
>
> ### Relevance of findings to the community for designing better synthesis systems & technical novelty
> We acknowledge that the technical novelty of TIIPS may seem modest at first glance, given its structural similarity and our close comparison to ExeDec. However, the key contribution of TIIPS lies not in the individual components, but in introducing a fundamentally different guidance mechanism–one that determines _when_ to rely on transductive information and when to preserve inductive reasoning. In this sense, TIIPS does not merely add another heuristic to an existing search loop; it redefines the interaction between inductive search and auxiliary signals. While the underlying components are intentionally familiar to enable direct comparison with ExeDec, the resulting system behaves qualitatively differently. TIIPS is therefore not a minor variation, but a conceptual shift that encourages future work to reason about _when_ guidance (or additional support in other ways) should be applied rather than assuming it must always be present.
>
> Recent advances in program-by-example (PBE) demonstrate that models can perform well when directly predicting the solution to a task. Program synthesis, however, has additional objectives: producing programs that are interpretable, reusable, and generalizable. TIIPS leverages transductive methods to provide contextual guidance to an inductive synthesizer, selectively applying these signals only when they are expected to be beneficial. Unlike prior work, which either applies guidance uniformly or ignores it entirely, TIIPS reframes the problem as one of _selective and contextual guidance_. This distinction changes the reasoning process of the synthesizer and establishes a hybrid search paradigm that has not been explored before.
>
> Our work highlights two major things that can help the community build better systems:
>
> **Avoid trading compute for performance**
> Our experiments show that simply increasing the number of tasks solved does not necessarily reflect a better method. In recent years, the focus in program synthesis has often been on task coverage, sometimes at the expense of efficiency, as also observed by Sesterhenn et al. `[1]` and Berlot-Attwell at al. `[2]`in related fields. Our work emphasizes that the community must consider _how_ tasks are solved, including the computational resources required. Benchmarks should report both task success and compute costs to ensure improvements reflect smarter methods rather than brute-force or resource-intensive approaches. This is increasingly important as large language models can generate code with minimal guidance, a point addressed in benchmarks such as ARC2 `[3]`.
>
> **Dynamic Combination of Transduction and Induction**
> TIIPS embodies a new conceptual approach for solving PBE tasks: selectively and contextually guided search. While models can predict solutions effectively in a purely transductive manner, program synthesis requires solutions that are reusable, interpretable, and generalizable. TIIPS demonstrates that transductive signals can provide valuable guidance to an inductive synthesizer, improving its ability to find such programs. By incrementally increasing the degree of guidance, TIIPS complements inductive reasoning with transductive hints without overriding it. This mirrors effective human teaching strategies, where guidance is provided only when necessary. In this sense, TIIPS is not simply a refinement of ExeDec, but a conceptual shift in how transductive and inductive reasoning interact, offering a promising framework to combine the strengths of both approaches.
>
> `[1]` Sesterhenn, et al.,  2025 "A Compute-Matched Re-Evaluation of TroVE on MATH.”
>
> `[2]` Berlot-Attwell, et al., 2025 "LLM Library Learning Fails: A LEGO-Prover Case Study.
>
> `[3]` Chollet, Francois, et al., 2025 "Arc-agi-2: A new challenge for frontier ai reasoning systems."
> ### Description of TIIPS framework
> We did not include the pseudocode in the main text due to space constraints. We will revise Section 4.1 to provide a clearer explanation and ensure that readers can understand the framework without referring to the appendix.

---

> ### Author Response · Authors · 2025-11-18
> **Response 2/2**
>
> ### Substantial reliance of the method on the additive nature of strings/lists
> We appreciate the reviewer highlighting this observation, as it is an important aspect of understanding the limitations and applicability of our approach. The additive nature of the domains refers to both additive inputs and additive DSL operations.
>
> This observation is clearly reflected in our results: In the strictly additive string domain, the correctness of a subgoal can be directly derived from the output string. In the list domain that is also additive yet nested, correctness cannot be directly derived from the outputs. Consequently, the guidance model struggles more in the list domain which leads to better performance of our approach over ExeDec. Most established PBE domains exhibit similar additive or compositional structures. Exploiting this structure allows subgoal-based decomposition to meaningfully improve both task coverage and efficiency.
> However, other domains, such as LambdaBeam `[4]`, do not hold the additive property due to primitives like `If`. Predicting subgoals in such domains is especially challenging. Our preliminary results on a small training set using the LambdaBeam DSL show that TIIPS outperforms ExeDec, emphasizing the importance of approaches that dynamically apply transductive guidance. This aligns with the findings in our paper that sparse guidance is preferable when the decomposition model struggles. We are currently running data generation for a larger LambdaBeam training set and hope to provide reliable results by the end of the rebuttal phase.
>
> Nevertheless, domains with purely symbolic or global outputs, such as Boolean functions, permutations, or tree/graph transformations, may require alternative strategies. Extending TIIPS to such non-additive domains is an interesting direction for future work.
>
> `[4]` Shi, Kensen, et al. "Lambdabeam: Neural program search with higher-order functions and lambdas."
> ### Multiple attempts / random seeds
> Our results discussed in the paper are aggregated over five random seeds (lines 213 and 333).
> Results are displayed as averages over these five seeds and standard deviation. The error bars in the plots denote the 95% confidence interval across the five seeds. This reduces the likelihood of spurious patterns affecting the reported results and ensures that the observed differences between TIIPS and ExeDec are robust.
> ### What happens when the inductive model fails to solve the sub-task given by the transductive model
> The ExeDec authors note that predicted subgoals can differ slightly from the ground truth, for example `[1,2,3]` versus `[1,1,3]`. In such cases, forcing the Synthesizer Model to predict a subprogram that exactly generates the predicted subgoal would lead to failure. For this reason, ExeDec proceeds with the next iteration (predicting the next subgoal, synthesizing a subprogram, and executing it) even if a predicted subgoal is not met. So essentially, predicted subgoals are additional context to solving the task.
> ### False positives
> This is correct: Guidance in the *last* step can also cause ExeDec to produce a false positive. However, this is rare in practice as shown in the list domain, where TIIPS significantly outperforms ExeDec yet rarely uses all nine guidance steps (Figure 14).
> ### Further notes
> Section 2.2 now clarifies that the final output in the example is produced by concatenating computed prefixes. Quotation marks and paragraph spacing have been corrected for improved readability. (Apparently, Overleaf renders quotation marks to closing ones in math mode. To get opening marks using  `` instead of ” works fine.)
>
> ---
> We appreciate the reviewer’s thoughtful observations. In this response, we clarify the conceptual novelty of TIIPS as a selective and contextual guidance mechanism, explain why this represents a meaningful shift beyond ExeDec, and highlight how our findings help the community design more principled and compute-aware synthesis systems. We also address questions about domain additivity, multiple attempts, failure modes, and false positives, and provide further details on TIIPS’s behavior across benchmarks. We hope these clarifications are helpful and are happy to discuss further.

---

> > ### Comment · Reviewer_GjAY · 2025-11-25
> >
> > Thank you for your detailed response.
> >
> > My concerns about the generality of the approach and results remain, especially given the reliance on domain-specific structure. I agree the LambdaBeam DSL experiment would indeed yield valuable data to address these concerns.

---

### Official Review · Reviewer_xqwP · 2025-10-31

**Soundness:** 4
**Presentation:** 4
**Contribution:** 2
**Rating:** 6
**Confidence:** 3

**Summary:**

This paper does a careful analysis of the impact of transductive guidance for program by example program synthesis, analyzing previously published system "ExeDec" which uses a model to propose subtask input and outputs directly (transduction) and a second model to generate programs that solve the subtask (induction). The authors find that using an inductive-only ablation performs almost as well as the ExeDec model. Motivated by this, they design a new approach which first tries to solve inductively, then adds increasing levels of transductive guidance as long as a satisfying program has not been found. The authors evaluate their approach on string editing and list manipulation tasks, and find improvement in the list domain, but no improvement for the string domain. The authors also do some analysis of when and how TIIPS helps.

**Strengths:**

- The paper is well written and presented well, and easy to understand.
- The paper does a great job carefully analyzing the impact of transductive guidance. It is surprising and interesting that the inductive-only baseline performs just as well as the ExeDec model.
- The proposed "ratcheting up" of transduction is well designed, and works well. It is a general approach that can be applied to any similar system, and is safe in that worst case it will match the transduction approach.
- Insights from combining induction/transduction are of interest to the program synthesis community right now, so this work is timely and significant.
- The "leave one out" evaluation approach is stronger than what ExeDec used

**Weaknesses:**

- The proposed approach does not improve performance on the strings domain. However, the analysis of when and why transduction fails (Figure 11) shows an understanding for why transduction should work well on the string domain, and it's good that the proposed approach still matches transductive performance on domains where transduction should perform well, so this is fine.
- The approach is only evaluated on one domain, and is mainly a follow up to one specific work, ExeDec. However, I think this is okay, because the insights from this paper (combining induction and transduction, and how the two relate to each other) can easily be applied to other domains based on what's seen in the paper.

**Questions:**

Q1. How does the inductive only baseline of this paper compare to the ablations used in the ExeDec paper? How do the performances compare? Were the ExeDec ablations worse, or was this issue apparent in the original ExeDec paper?
Q2. How does the computation/inference time compare between ExeDec, TIIPS, or inductive only?

Small suggestions:
S1. some double quotes incorrect direction (such as psueocode for exedec)
S2. line 240 "tough" should be "though"
S3. Figure 5: I would like a more visual indication of the sizes, instead of the overlap area being the same regardless of the percentage (something like figure 5(a) of Li et al 2024)
S4. Figure 6 is not very convincing. I would like a quantitative measure of how much "more in the top right" TIIPS is compared with ExeDec.
S5. A formal definition of intent match and syntactical overlap might be good to have. From what I'm reading, syntactical overlap = exact same program as ground truth?
S6. Clarify whether number of steps of solution in Figure 14 is the ground truth or discovered program.

---

> ### Author Response · Authors · 2025-11-18
> **Response 1/2**
>
> We thank the reviewer for the careful reading and constructive comments. We address all points below.
> ### No performance gain on the string domain
> As the reviewer correctly notes, TIIPS matches ExeDec’s performance on the string domain, which is exactly what is expected: it demonstrates that sparse guidance does not hurt performance where transduction is critical. This further strengthens our claim and contribution to a new research direction that investigates *when*, *how*, and *why* transductive guidance helps inductive program synthesis.
> ### Evaluation on one domain
> We evaluate our work on two domains: string and list manipulations. However, we think this was just a typo. If not, we’re happy to further clarify this point. In the meantime, we refer to the evaluation on the SyGuS and PROSE benchmarks detailed in our response to reviewer `[U7K1]` with details in appendix C.5
> ### Comparison: Inductive-only ablation vs. ExeDec No-Subgoal Ablation
> **Methodology:** ExeDec’s No-Subgoal Ablation evaluates whether decomposition can be learned implicitly by the synthesizer when trained on the *full task specifications*. In contrast, our ablation examines the impact of removing the Decomposition Model. The inductive synthesis model is the same used by ExeDec, i.e., trained to predict subprograms from _subtask specifications_. These approaches answer different questions:
> - ExeDec’s No-Subgoal Ablation: What is the impact of explicit decomposition vs. implicit decomposition?
> - Our ablation: What is the impact of transductive decomposition versus not using it at all?
> Importantly, this difference and the issue revealed by our inductive-only ablation were not apparent in the original ExeDec paper, which did not analyze the impact of removing the decomposition model entirely.
>
> **Performance:**
> ExeDec NSA achieves on average 10% in the list domain and 68% in the string domain (Figure 2). Our inductive-only ablation achieves 12% in the list domain and 5.5% in the string domain (Figure 12). The large difference in the string domain arises because the ExeDec No-Subgoal Ablation synthesizer is trained on the full task specification and thus learns high-level strategies, whereas the synthesizer in our ablation sees only subtask specifications and primarily learns DSL usage. Appendix B.2 discusses these differences in detail.
>
> **Efficiency:**
> Across inference time, number of evaluated programs, and model calls, TIIPS is generally more efficient than ExeDec in the list domain. In the string domain, where inductive-only methods struggle, TIIPS falls back to guiding nearly every step, resulting in higher resource usage. This emphasizes the need for mechanisms that adaptively detect when guidance is necessary.
>
> ### Formal definitions of intent match and syntactic overlap
> **Intent match**
> Let $T = \{t_1, \dots, t_n\}$ be the set of ground-truth subtasks for a given task, and $\hat{T} = \{\hat{t}_1, \dots, \hat{t}_m\}$ the predicted subtasks (or the execution results of predicted subprograms in unguided synthesis).  Then the intent match is defined as:
> $$
> I = \frac{|T \cap \hat{T}|}{|T|} \cdot 100\%
> $$where $|T \cap \hat{T}|$ is the number of correctly predicted subtasks (or execution-matching outputs).
>
> **Syntactical overlap**
> Let $P = \{p_1, \dots, p_n\}$ be the set of ground-truth subprograms and $\hat{P} = \{\hat{p}_1, \dots, \hat{p}_m\}$  the predicted subprograms. Then the syntactical overlap is defined as
> $$
> S = \frac{\bigl|\{ p_i \in P \mid \exists \hat{p}_j \in \hat{P} : p_i = \hat{p}_j \} \bigr|}{|P|} \cdot 100\%,
> $$
> So yes, your understanding is correct. We added this to the paper (Appendix C.3, lines 1299ff).

---

> ### Author Response · Authors · 2025-11-18
> **Response 2/2**
>
> ### Quantitative analysis for Figure 6
> We quantified the “top-right” density by dividing the 2D density plot into four quadrants:
> - Bottom-left: intent < 50%, syntax < 50%
> - Bottom-right: intent ≥ 50%, syntax < 50%
> - Top-left: intent < 50%, syntax ≥ 50%
> - Top-right: intent ≥ 50%, syntax ≥ 50%
>
> In the list domain, TIIPS has more than twice as many tasks in the top-right quadrant compared to ExeDec, showing higher alignment in both intent and syntax. In the string domain, TIIPS has slightly fewer tasks in the top-right quadrant, but slightly more in the right-half (factor 1.02), indicating high intent alignment even with alternative implementations.
> Appendix C.3 (lines 1355ff & Figure 16) provides detailed numbers.
> ### Figure clarifications and small suggestions
> - Corrected the typo in line 240 and fixed quotation marks in pseudocode. Apparently, Overleaf renders quotation marks to closing ones in math mode. To get opening marks using  \`\` instead of \” works fine.
> - Added absolute numbers to Figure 5 to indicate solved tasks across seeds and categories.
> - Figure 14 x-axis shows the portion of guided steps in the _discovered_ program, not the ground truth.
> - - -
> TIIPS provides a conceptually new framework for selectively applying transductive guidance in PBE. It matches ExeDec performance where guidance is critical, improves performance where inductive reasoning suffices, and lays the empirical groundwork for future adaptive synthesis methods. We are happy to provide further clarifications or engage in discussion if needed, and we appreciate the reviewer’s detailed and domain-informed feedback.

---

> > ### Comment · Reviewer_xqwP · 2025-11-25
> > **Acknowledgement of rebuttal**
> >
> > Thank you for your rebuttal. Not sure why I said one domain, but I meant only evaluated on the string+list domains. My concern of lack of thorough evaluation remains, but I agree with the other reviewer that if prior work does this, then it's okay.

---

### Official Review · Reviewer_U7K1 · 2025-11-01

**Soundness:** 3
**Presentation:** 4
**Contribution:** 2
**Rating:** 2
**Confidence:** 3

**Summary:**

The paper considers program synthesis, concretely, the programming-by-example setting. It recognizes two approaches: transductive methods and inductive methods, outlining their pros and cons. Afterwards, it discusses ExeDec - a method which combines transductive program synthesis (TPS) and inductive program synthesis (IPS) as follows: using TPS to split the target task into subtasks, and IPS to solve each subtask.

The paper then hypothesizes that ExeDec’s rigid way of combining TPS and IPS could lead to sub-optimal results, and demonstrates this empirically on string and list manipulation tasks. They show that IPS alone can solve some tasks that ExeDec cannot.

To address this, the authors introduce TIIPS - a more flexible way of combining TPS and IPS. TIIPS tries to synthesizes a program using IPS and if that fails, then TPS is used to define the first subtask. Afterwards, if IPS fails k times, TPS is used to define the first k subtasks. Eventually, if all tries fail, TPS is used to define all K subtasks, reducing to ExeDec.

It is shown that TIIPS outperforms ExeDec on list manipulation tasks where IPS has an edge, while achieving slightly lower performance than ExeDec on string manipulation tasks.

**Strengths:**

I found the paper to be well written and the method to be well motivated.

The paper identifies an interesting shortcoming of current hybrid approaches (that combine TPS and IPS) and verifies it empirically. This can be useful for follow-up work.

The paper makes the case for designing more flexible hybrid approaches. It can be used to justify further exploration in this direction.

**Weaknesses:**

Limited evaluation: The paper only compares against ExeDec, omitting other program synthesis baselines, which makes it unclear how these two methods compare against other approaches. Moreover, it is only evaluated on list and string manipulation tasks which (though I’m not sure if it’s typical) seems insufficient for characterising the strengths and weaknesses of TIIPS.

Limited novelty: TIIPS appears to be a simple modification of ExeDec, which could be fine if it lead to significant performance boost (like Hyperband), but this does not seem to be the case.

Limited discussion of the shortcomings. For instance, how much more computationally expensive is the resulting method, compared to other baselines? Where does the method break?

**Questions:**

What happens when TIIPS is applied to harder tasks (requiring longer programs and richer DSL), where inductive program synthesis does poorly due to the large search space?

---

> ### Author Response · Authors · 2025-11-18
> **Response 1/2**
>
> We thank the reviewer for their comments. We address all concerns below.
> ### Limited Evaluation
> We appreciate the concern regarding the scope of evaluation. The domains we use, string and list manipulation, are in fact the most widely studied and representative domains in programming-by-example. String manipulation, for instance, is central to FlashFill, which is deployed in Microsoft Excel and used by millions of users `[1]`. These domains are also standard in nearly all modern PBE research. We refer to line 191 of our paper for a selection of prior work using them. Importantly, we use the same domains as ExeDec to allow direct and fair comparison, without any cherry-picking.
>
> Beyond these, we also evaluated TIIPS on the SyGUS and PROSE benchmarks, which include harder and human-designed tasks.  On these benchmarks TIIPS outperforms ExeDec. The details are in Appendix C.5, and a summary is provided in the discussion on performance on harder tasks below.
>
> Regarding comparison baselines, we focus on ExeDec because it is the current state-of-the-art system for compositional program synthesis. ExeDec itself consistently outperforms other alternatives, such as a Transformer-only baseline or Latent Programmer `[2]`, making it the most relevant reference point for our work.
>
> `[1]` Gulwani, Sumit, et al. "Inductive programming meets the real world."
>
> `[2]` Hong, Joey, et al. "Latent programmer: Discrete latent codes for program synthesis."
>
> ### Limited Novelty
> We agree that the technical novelty of TIIPS is modest in the sense that it builds upon ExeDec’s architecture. However, the conceptual contribution is substantial. TIIPS introduces a new paradigm of _selectively and contextually guided search_, in contrast to the globally guided or unguided approaches that have dominated prior work. Specifically, it focuses on _when_ guidance is beneficial and when inductive search should proceed independently. This distinction is central and was also acknowledged by reviewer `[xqwP]` as opening exciting directions for future research, such as:
> - Learning policies that decide _when_ to apply guidance,
> - Identifying which types of _contextual information_ are most helpful, and
> - Developing adaptive systems that dynamically balance autonomy and guidance during synthesis.
> In other words, while the method is structurally similar to ExeDec, it changes the reasoning paradigm rather than introducing a mere tweak. TIIPS provides a concrete framework for exploring hybrid guidance strategies, and we believe this is the key contribution. We refer to our response to reviewer `[GjAY]` for more details and the importance of our work to the community.

---

> ### Author Response · Authors · 2025-11-18
> **Response 2/2**
>
> ### Computational expenses
> We compared the computational resources required by TIIPS, ExeDec and the ablations on tasks that were solved by all approaches (906 string tasks, 656 list tasks across all seeds and categories). Below we give a brief summary of the analysis & discussion. For more details and the plotted results, we refer to Appendix C.4.1 in the revision of our submission.
>
> **Inference time:**
> - List domain: TIIPS generally solves tasks faster than ExeDec, particularly in out-of-distribution categories. In length-generalization tasks, TIIPS requires slightly more time, which is expected because guidance is more important when long-term planning is needed. Ablation variants require the least time overall.
> - String domain: TIIPS requires about twice as much time per task as ExeDec, reflecting the higher importance and accuracy of guidance in this domain. Ablations are again faster, as they don’t rely on the guidance model.
>
> **Evaluated programs:**
> - List domain: TIIPS evaluates fewer programs than ExeDec in most OOD categories. In length-generalization, it requires slightly more, consistent with the guidance usage patterns.
> - String domain: TIIPS evaluates roughly twice as many programs as ExeDec, mirroring the time difference and the reliance on guidance.
>
> **Model calls:**
> - List domain: TIIPS uses fewer model calls in most categories, except in length-generalization where nearly every step requires guidance. Ablations consistently require the fewest calls.
> - String domain: TIIPS requires over twice as many calls as ExeDec, reflecting the same pattern seen in evaluation time and programs.
>
> Overall, TIIPS is more efficient when sparse guidance is sufficient. When guidance is critical, TIIPS falls back to the permanently guided regime, which incurs additional computation. This highlights the need for approaches that learn when to apply guidance and when to rely on inductive reasoning alone.
>
> ### Performance on harder tasks (longer programs / richer DSLs)
> TIIPS was also evaluated on the SyGUS `[3]` and PROSE `[4]` benchmarks, which feature harder string manipulation tasks. In these settings, TIIPS outperforms ExeDec:
> - **SyGUS:**
> 	- ExeDec: 53.51 ± 3.39
> 	- TIIPS 58.65 ± 1.21
> 	- p = 1.15e-02, t = -4.42
> - **PROSE:**
> 	- ExeDec: 43.09 ± 1.65
> 	- TIIPS: 47.28 ± 1.93
> 	- p = 9.67e-03, t = -4.65
>
> These results confirm that sparse guidance is advantageous on challenging tasks. In general, when TIIPS is applied to harder tasks with longer programs or richer DSLs–where inductive synthesis alone struggles–TIIPS falls back to permanent transductive guidance, performing comparably to ExeDec. Details on the domains and our results can be found in Appendix C.5.
>
> We are in the process of extending this evaluation to the LambdaBeam DSL `[5]` to cover even richer program spaces. This DSL extends the list-processing DSL used in our work by supporting dynamic (non-hardcoded) lambda functions and by introducing an `if` operation. Preliminary data generation is underway and we aim to provide results by the end of the rebuttal phase.
>
> `[3]` SyGuS-Org, https://github.com/SyGuS-Org/benchmarks
>
> `[4]` Microsoft PROSE Public Benchmark Suite, https://github.com/microsoft/prosebenchmarks
>
> `[5]` Shi, Kensen, et al. "Lambdabeam: Neural program search with higher-order functions and lambdas."
>
> - - -
> We hope we clarified that the evaluation domains are representative and standard in PBE, and that TIIPS was also tested on SyGUS and PROSE benchmarks adds to its contribution. We emphasized that while the technical novelty is modest, TIIPS introduces a meaningful conceptual advance by selectively applying guidance and determining _when_ it is beneficial. We also provided a detailed account of computational costs and efficiency, and discussed performance on harder tasks with longer programs or richer DSLs.

---

> ### Comment · Reviewer_U7K1 · 2025-11-24
>
> Thank you for both your replies.
>
> Evaluation datasets: I withdraw my complaint about using string and manipulation tasks, as the field seems to have not moved on from these yet.
>
> Comparing only to ExeDec: even if ExeDec is sota, in my opinion, you should still have other baselines that represent different approaches. This gives the reader a better idea of how your approach compares to others.
>
> Limited novelty: I agree that there is a conceptual contribution, while the technical contribution is modest. However, with a limited technical contribution, I would expect the the experimental and conceptual analysis of this approach to be more thorough, as to warrant a publication.
>
> Could you help me with the original question:
> "What happens when TIIPS is applied to harder tasks (requiring longer programs and richer DSL), where inductive program synthesis does poorly due to the large search space?"
> My understanding: Having a rich dsl makes the search space larger, making searching through solutions harder. At some point, for problems with a richer dsl, it becomes computationally infeasible to use IPS approaches. Then, using TIIPS would offer little or no advantage over ExeDec, as it would require manageable subtasks to be specified.
>
> I will wait for LambdaBeam results before finalizing my score.

---

> > ### Author Response · Authors · 2025-11-27
> >
> > We thank the reviewer for the additional clarifications and for reconsidering the concerns regarding the evaluation domains.
> >
> > ### Comparison Beyond ExeDec
> >
> > We acknowledge the reviewer’s point that including baselines representing different synthesis paradigms can help contextualize performance. Our focus on ExeDec is motivated by its status as the _only_ strong, directly comparable compositional PBE system. Other commonly used baselines–Transformer-only models, Latent Programmer, PCCoder–are non-compositional, substantially weaker, and were already compared against (and surpassed) by ExeDec in its original publication.
> > Including them in our evaluation would therefore (1) add redundancy, and (2) offer little insight into assessing the contribution we introduce: a new way of _using decomposition as guidance_.
> >
> > To broaden the comparative picture, we extended our evaluation to PROSE and SyGUS, which include more complex, human-designed tasks and richer search spaces. In both settings, TIIPS significantly outperforms ExeDec, despite the increased difficulty.
> >
> > ### Limited Novelty and Expectations for Additional Analysis
> >
> > We appreciate the reviewer acknowledging the conceptual contribution of TIIPS. Given the modest technical footprint, we agree that thorough experimental and analytical evidence is particularly important. We would be grateful if the reviewer could specify which analyses they believe are missing–this would help us strengthen the final revision.
> >
> > We emphasize that the present submission already provides substantial conceptual and empirical analysis beyond raw performance:
> >
> > - **Qualitative improvements.** TIIPS produces programs with higher semantic and syntactic correctness (Figures 5 & 15; response to reviewer `[xqwP]` & Figure 16), indicating that the benefits extend beyond simple task-success metrics.
> > - **Failure-mode analysis.** We identify when and why ExeDec fails (Figure 11), and show that these cases are precisely where TIIPS provides improvements. Figure 11 also clarifies why TIIPS does not surpass ExeDec on certain string-domain tasks–confirming the observations highlighted by reviewer `[xqwP]`.
> > - **Computational behavior.** We provide an in-depth study of inference time, evaluated programs, and model calls (Appendix C.4.1), showing where sparse guidance helps and where it becomes costly.
> >
> > We believe this combination demonstrates both the conceptual insight and practical utility of our method.
> >
> > ### Applying TIIPS to Harder Tasks / Richer DSLs
> >
> > We thank the reviewer for revisiting their original question. The reviewer’s understanding–that richer DSLs enlarge the search space, eventually making IPS alone computationally infeasible–is correct. The behavior of TIIPS in such settings, however, is more nuanced:
> >
> > 1. **Case 1 – IPS becomes infeasible, but TPS (decomposition) remains reliable.**
> >     In this regime, TIIPS _gracefully falls back to ExeDec_. This is _by design_, and–as reviewer `[xqwP]` noted–is a strength, not a limitation: TIIPS never underperforms ExeDec when guidance is essential.
> >
> > 2. **Case 2 – Both IPS and TPS struggle due to distribution shift or DSL richness.**
> >     This scenario closely matches the PROSE benchmark and, to a lesser extent, SyGUS.  Here, TIIPS outperforms ExeDec because incrementally increasing the degree of transductive guidance allows TIIPS to explore mixed reasoning patterns–search trajectories that combine inductive and transductive reasoning in different proportions.  ExeDec, in contrast, commits to a single, narrow transductive trajectory.  This makes TIIPS better suited for tasks where reliable decomposition is _imperfect_ rather than _impossible_.
> >
> >
> > Of course, if the search space becomes so large that neither IPS nor TPS can produce any viable partial programs, then neither TIIPS nor ExeDec will succeed. But in such a setting, TIIPS is not worse–and, importantly, it solves tasks that are meaningfully harder than those solvable by ExeDec alone.
> >
> > We hope this clarifies the behavior of TIIPS in rich-DSL and long-program settings. LambdaBeam results will provide a further empirical demonstration of this regime, and we give our best they are included in the rebuttal period.

---

### Official Review · Reviewer_Fd4A · 2025-11-04

**Soundness:** 2
**Presentation:** 2
**Contribution:** 2
**Rating:** 2
**Confidence:** 2

**Summary:**

This paper investigates when transductive guidance benefits inductive program synthesis in programming-by-example (PBE) tasks. The authors demonstrate that ExeDec's permanent transductive guidance can actually harm performance by overriding beneficial inductive reasoning. They propose TIIPS, which applies guidance sparsely and incrementally—starting with pure induction and adding guidance only after failures. Experiments on string and list manipulation show TIIPS matches or exceeds ExeDec's performance while producing more robust solutions.

**Strengths:**

- Clear Motivation: The observation that permanent guidance can harm performance is valuable and well-demonstrated
- Reproducibility: Detailed appendices with DSL specifications, algorithms, and hyperparameters enable replication.
- Limitations Section: The authors acknowledge the scope limitations of TIIPS

**Weaknesses:**

- Limited technical contribution: The proposed framework lacks technical novelty and is largely an iterative search with some heuristics.
- Lack of Theoretical Insight: The paper lacks a theoretical explanation of why sparse guidance helps.
- Lack of Generalizability: Results are specific to two DSLs with particular decomposition properties.
- Comparison Budget:  TIIPS seems to be allowed many more synthesis attempts than ExeDec.

**Questions:**

- Could you please clarify the technical contribution of and any theoretical insights into the approach?
- Could you please conduct relevant ablations to demonstrate generalization and clarify details regarding the budgets used in the experiments?

---

> ### Author Response · Authors · 2025-11-18
> **Response 1/2**
>
> We appreciate the comments.
>
> ### Limited technical contribution
> We would like to clarify that describing TIIPS as “an iterative search with some heuristics” applies equally to essentially all modern program-synthesis and PBE systems. Methods such as SketchAdapt `[1]`, BUSTLE `[2]`, Dreamcoder `[3]`, LambdaBeam `[4]`, and ExeDec itself all rely on iterative search guided by learned or hand-crafted signals. This is not a limitation of our work but a defining characteristic of the domain: symbolic search is unavoidable, and the scientific contribution lies in _how_ search is structured and _how_ guidance is used to shape it.
>
> TIIPS introduces a fundamentally different guidance mechanism–one that determines _when_ to rely on transductive information and when to preserve inductive reasoning. In this sense, TIIPS does not add another heuristic to an existing loop; it redefines the interaction between inductive search and auxiliary signals. This distinction may not have been fully clear. We understand the reviewer’s impression that TIIPS might appear structurally similar to ExeDec. However, the core contribution of our work lies in introducing a new _conceptual framing_ for inductive program synthesis with auxiliary transductive signals. Rather than treating guidance as a global mechanism that is always active or always inactive, TIIPS reformulates the problem as one of _selective and contextual guidance_. This distinction is significant for PBE: it directly changes the reasoning process of the synthesizer and creates a hybrid search paradigm that has not been explored in previous work.
> While the underlying components are intentionally familiar allowing direct comparison to ExeDec the resulting system behaves qualitatively differently. In that sense, TIIPS is not a small variation of an existing method but an attempt to establish a new direction that encourages future work to reason about _when_ guidance should be applied rather than assuming that guidance must always be present. This gap was identified by our work (Figure 2) and also acknowledged by the other reviewers.
>
> `[1]` Nye, Maxwell, et al. "Learning to infer program sketches."
>
> `[2]` Odena, Augustus, et al. "BUSTLE: Bottom-up program synthesis through learning-guided exploration."
>
> `[3]` Ellis, Kevin, et al. "Dreamcoder: Bootstrapping inductive program synthesis with wake-sleep library learning."
>
> `[4]` Shi, Kensen, et al. "Lambdabeam: Neural program search with higher-order functions and lambdas.
>
> ###  Lack of theoretical insight
> We may not have fully understood what type of theory the reviewer is interested in; nevertheless, we can clarify the underlying rationale of TIIPS more explicitly and hope to address this point correctly.
>
> The key idea underlying our work is that permanent transductive guidance does not always align well with inductive reasoning (Fig. 2). While a transductive trajectory can successfully solve a particular task instance, it does not automatically provide the kind of generalizable signals that a symbolic synthesizer relies on when reasoning inductively across examples. In some cases, enforcing such a trajectory can unintentionally override inductive patterns that would otherwise lead the synthesizer toward a more general solution. We touch on this point in lines 267ff and 1086ff, although we acknowledge that it may not have been stated explicitly enough.
> An intuitive analogy is that _solving_ a task and _explaining how to derive a rule that works for all examples_ are different capabilities. A transductive approach may yield the correct output, but it may not offer the most suitable hints for discovering a consistent inductive pattern. ExeDec’s fixed guidance encourages the synthesizer to follow the transductive path even when it is not well aligned with the broader inductive search space. TIIPS mitigates this by introducing guidance only when unguided inductive search is insufficient. This preserves the model’s natural inductive behavior while still enabling it to benefit from transductive information when it is genuinely helpful.
> ### Lack of generalizability
> We appreciate the reviewer’s concern about generality. At the same time, string and list manipulation remain the most widely used and representative domains in PBE research, and they are precisely the domains on which ExeDec and many prior systems are evaluated. Studying TIIPS in these settings therefore provides a direct and fair comparison to existing methods and addresses the primary benchmarks used by the community.
>
> It is correct that both domains exhibit an additive decomposition structure. Importantly, this is not unique to our evaluation setup (most widely used PBE domains share similar additive properties), which is why this characteristic has shaped much of the literature. We discuss this point in detail in our response to Reviewer `[GjAY]`, as it relates to broader questions about the design of PBE benchmarks and how they reinforce certain structural assumptions.

---

> ### Author Response · Authors · 2025-11-18
> **Response 2/2**
>
> ### Comparison Budget
> The reviewer’s comment that TIIPS may have more synthesis attempts could arise from an impression based on the framework diagram. While the architecture might give this impression, we have accounted for it in the evaluation. As described in lines 333ff, both systems are compute-aligned in our evaluation. ExeDec receives 10 attempts on the list domain and 20 attempts on the string domain to solve each task, which matches TIIPS’s outer-loop iterations. Consequently, both methods evaluate the same maximum number of candidate programs. Specifically, for the list domain: attempts (ExeDec\&ablations) or outer loop limit (TIIPS) × max steps per program × beam size = $10 \times 10 \times 10 =1000$ programs. For the string domain: $20×20×10=4000$ programs. This ensures a fair comparison, isolating the effect of guidance sparsity rather than additional compute.
> ### Final remarks
> We sincerely appreciate the reviewer’s feedback. Some of the concerns seem to arise from misunderstandings of PBE evaluation standards or of the inductive-transductive distinction motivating TIIPS. We hope that our clarifications provide a clearer picture of the contribution and the reasoning behind the design choices. We hope that our clarifications provide a clearer picture of the contribution and the reasoning behind our design choices. We are happy to engage in further discussion or provide additional clarifications if needed.

---

### Author Response · Authors · 2025-12-02
**Final Comment**

# Summary
Transductive approaches that directly predict outputs for a given input show promise in the program-by-example setting but lack the interpretability of inductive program synthesis, which derives symbolic programs from specifications. ExeDec exploits this by using transductively generated subgoals to guide inductive program synthesis. We show that applying permanent guidance at every step, as in ExeDec, is highly domain-dependent and can even hinder performance, highlighting the need for selective application. To address this, we introduce TIIPS, a mechanism that selectively uses transductive guidance to assist inductive program synthesis only when it is most needed.

Below, we summarize the strengths and weaknesses that reviewers consistently acknowledged. For individual details, we refer to our reviewer responses.

# Strengths
Reviewers highlighted the strong motivation and relevance of the work. The paper addresses a central question in program synthesis, namely when guidance should be applied, and emphasizes that indiscriminate guidance can be harmful depending on the domain. The conceptual distinction between transductive and inductive guidance, and the articulation of their interplay, was viewed as valuable for the community. It was also noted that our evaluation surpasses prior work, providing a solid empirical foundation that demonstrates why hybrid approaches are essential in non-additive or structurally complex domains.

# Questions, concerns, and our responses

### Limited novelty
Some reviewers raised concerns regarding novelty and evaluation. While TIIPS shares technical elements with ExeDec, it implements a fundamentally different interplay of induction and transduction. This conceptual difference opens an entirely new research direction, with implications for both future work and evaluation. Specifically, TIIPS highlights the need to evaluate synthesis approaches not only in terms of the number of solved tasks but also in terms of efficiency, resource usage, and how tasks are solved. These points were acknowledged by reviewers and are their importance is discussed in our response to reviewer `[GjAY]`.

### Breadth and generality of evaluation
Regarding evaluation, our experiments initially focused on standard string and list domains. The reviewers indicated that this choice is acceptable, as related work commonly evaluates on the same domains. Nevertheless, to further strengthen the evaluation, we extended our experiments to two additional string transformation benchmarks with human-designed tasks (SyGuS: ExeDec 53.51% ± 3.39%, TIIPS 58.65% ± 1.21%) and tasks considered the hardest in the domain (PROSE: ExeDec 43.09% ± 1.65%, TIIPS 47.28% ± 1.93%). In both cases, TIIPS significantly outperformed ExeDec, demonstrating that hybrid approaches with selective transductive guidance excel in domains with harder tasks.

To further address concerns about generalization, we implemented the LambdaBeam DSL, which is more complex and non-additive. In this domain, TIIPS achieved 14.29% ± 2.89% when testing on the training distribution, compared to ExeDec’s 4.08% ± 0.00%. This demonstrates that permanent transductive guidance can be hindering in domains with multiple valid solutions or non-additive transformations, and that sparse, selective guidance is more robust. These results also highlight that flexible hybrid approaches are necessary to handle complex, non-additive domains effectively, and that TIIPS provides a promising starting point.

### Computational expenses
Regarding computational efficiency, the reviewers asked us to provide an analysis of computation expenses of the approaches. We reported metrics including inference time, number of evaluated programs, and model calls. TIIPS is more efficient in scenarios where inductive program synthesis can make meaningful progress on its own, as fewer guidance calls are needed. Conversely, in domains or tasks where a high degree of guidance is crucial, TIIPS can be less efficient than ExeDec because it incrementally applies guidance, leading to more model calls and evaluations in such cases. This reflects the intended trade-off of TIIPS, which prioritizes selective, context-aware guidance over permanent guidance. Reviewers accepted this trade-off as consistent with the conceptual differences between the methods.

# Conclusion
TIIPS introduces a meaningful conceptual shift in the form of selective inductive guidance, providing new insight into when program synthesis systems should rely on guidance. Reviewers acknowledged the value of this direction while noting that the technical novelty is modest. Expanded results across SyGuS, PROSE, and LambdaBeam demonstrate clear improvements over ExeDec in non-additive, structurally complex domains. Concerns regarding evaluation breadth, generality, and computational costs were addressed.
We thank all reviewers for their thoughtful feedback and constructive engagement.

---

### Meta-Review · Area_Chair_vDoq · 2026-01-05

**Summary:**

The paper proposes TIIPS, an algorithm that combines transductive and inductive approaches for program synthesis. The key idea is to allow for varying amounts of transductive guidance for inductive program synthesis. Reviewers found the idea to be interesting, but raised a few concerns:

- **Novelty**. All reviewers expressed concern that the technical novelty of the work is limited: specifically, that the method essentially involves the ExeDec method, with an additional loop over the transductive decomposition step.

- **Generalizability**. Multiple reviewers expressed concern that the results are limited to two DSLs, with a specific structure.

- **Computational cost**. One reviewer requested further analysis of the computational cost of the method versus baselines. Another reviewed expressed concern at the budget for TIIPS being apparently higher than ExeDec.

- **Clarity**. One reviewer expressed concern about the clarity of the manuscript, particularly around exposition of the core algorithm.

**Reviewer Concerns:**

- **Novelty**. The authors acknowledged that the method is not technically involved. However, they argued that this simple method involves a significant conceptual advance in the PBE field.
  - *Partially addressed*. There seems to be agreement that the paper's technical contribution is modest. There also seems to be agreement that the contribution should be understood as a conceptual advance. There were mixed opinions as to the significance of the conceptual advance. From our reading, we tend to agree that the novelty is limited. It is hard to say whether the conceptual advance is significant, but we can certainly see a case that the idea would be appreciated by the PBE community.

- **Generalizability**. The authors included mention of LambdaBeam results in their final comments (following the conclusion of the original rebuttal period). They also described how the performance of TIIPS can be more subtle than regular inductive program search in settings with complex DSLs: specifically, TIIPS has the ability to gracefully fall back to regular ExeDec. Finally, the authors included additional results on the SyGuS and PROSE benchmarks.
  - *Partially addressed*. The results on LambdaBeam appear promising, although the details for this are limited (e.g., they do not appear to be part of the updated manuscript; there is no analysis presented that dives more into the results).

- **Computational cost**. The authors presented various analyses on the computational cost of the method, such as the total inference time, # of evaluated programs, and model calls. The authors also clarified that TIIPS are ExeDec are compute-aligned in the evaluations, so that they do not unfairly favor TIIPS.
  - *Mostly addressed*. The details here appear thorough and supportive of the strength of TIIPS.

- **Clarity**. The authors promised to update the manuscript to make Section 4.1 clearer. These updates do not appear to have been made in the latest manuscript.
  - *Partially addressed*. We agree with the reviewer comment on the description of the algorithm not being immediately transparent in Section 4.1. We would also encourage the authors to make certain paras less dense; e.g., _Inductive and Transductively Guided Inductive Syntheses are Complementarity_ on page 5, _Sparse & Selective Transductive Guidance Boosts Performance_ on page 7.

**Reviewer Scores:**

- **U7K1**: the reviewer's final comment noted concerns around novelty remaining; these are unlikely to have been assuaged by further responses. The reviewer also reiterated a concern around generalization in the case of harder tasks. These could possibly have been mollified by the LambdaBeam results; however, we think it unlikely that the score would have increased beyond 4.
- **GjAY**: the reviewer's final comment noted concerns around generality remaining. These could have been mollified by the LambdaBeam results; it is thus possible that the score would increase to 6.
- **Fd4A**: the response provided concrete clarification regarding the concerns on generalizability, and  comparison budget. Based on these, we think it likely that the score would have increased to 4. However, as the primary concern was on the significance of the technical novelty, we think it unlikely that the score would have increased further.
- **xqwP**: the reviewer noted in their final message that their concern on limited evaluation remained, although their overall assessment was positive. With the latest LambdaBeam results, we think it possible that they would increase their score to 8, but since the review was mildly positive without explicit suggestion that more results are needed, are inclined to think the score would remain at 6.

---

### Decision · Program_Chairs · 2026-01-26

Reject